



# Retrieval of process rate parameters in the general dynamic equation for aerosols using Bayesian state estimation

Matthew Ozon[1], Aku Seppänen[1], Jari P. Kaipio[1,3], and Kari E.J. Lehtinen[1,2]

[1]Department of Applied Physics, University of Eastern Finland, Kuopio, Finland
[2]Finnish Meteorological Institute, Kuopio, Finland
[3]Department of Mathematics, Faculty of Science, University of Auckland, New Zealand

**Correspondence:** matthew ozon (matthew.ozon@uef.fi)

**Abstract.** The uncertainty in the radiative forcing caused by aerosols and its effect on the climate change calls for research to improve knowledge of the aerosol particle formation and growth processes. While the experimental research has provided large amount of high quality data on aerosols in the last two decades, the inference of the process rates is still inadequate, mainly due to limitations in the analysis of data. This paper focuses on developing computational methods to infer aerosol process rates

from size distribution measurements. In the proposed approach, the temporal evolution of aerosol size distributions is modeled with the general dynamic equation equipped with stochastic terms that account for the uncertainties of the process rates. The time-dependent particle size distribution and the rates of the underlying formation and growth processes are reconstructed based on time series of particle analyzer data using Bayesian state estimation – which not only provides (point) estimates for the process rates but also enables quantifying their uncertainties. The feasibility of the proposed computational framework is

demonstrated by a set of numerical simulation studies.

## 1 Introduction

Aerosols scatter and absorb solar radiation and affect the permeability of the atmosphere to solar energy (the direct effect). In addition, aerosol particles act as seeds for cloud droplets (cloud condensation nuclei, CCN) and thus influence the properties of clouds (the indirect effect, for example Ramanathan et al. (2001)). Worldwide, particulate air pollutants are also responsible

for up to 7 million premature deaths per year (WHO 2014). The Intergovernmental Panel on Climate Change (IPCC, 2013 Stocker (2014) and 2014 Pachauri et al. (2014)) recognized the uncertainty in the radiative forcing caused by aerosols as the main individual factor limiting the scientific understanding of future and past climate changes.

Some of the uncertainty is caused by the fact that the initial stages of the new particle formation (NPF) processes in the atmosphere are still not completely known. It has been known for close to a hundred years that photochemically driven NPF

may occur in the atmosphere Aitken (1889). The fact that it occurs regularly throughout the troposphere has, however, become clear only during the last fifteen years or so Kulmala et al. (2004). Partly because of this, the systematic development of parameterizations describing tropospheric NPF as an active research topic has only just begun. Studies with these models suggest that tropospheric NPF may have significant effects on cloud condensation nuclei (CCN), and thus also, the global cloud albedo effects of atmospheric aerosols overall Merikanto et al. (2010). One challenge in estimating the anthropogenic



aerosol effect on climate is the need to know the preindustrial conditions and dynamics, which is the baseline of forcing
estimations Hansen et al. (1981). Very recently, Gordon et al. Gordon et al. (2016) made new estimations of anthropogenic
aerosol radiative forcing by assuming pure biogenic particle formation, as suggested by Kirkby et al. Kirkby et al. (2016) based
on the CLOUD-experiments at CERN. The increased particle formation rates at the preindustrial conditions increased the CCN
conditions and further, the cloud albedo — resulting in a 0.22 W/m2 decrease in anthropogenic aerosol forcing. As the NPF
treatment in global climate models is typically based on parameterizing particle formation and growth rates from chamber
experiments such as CLOUD or from field measurements, it is of great importance to analyze the data with care, paying also
attention to the uncertainties.

     Most of the research dealing with NPF event analysis has used similar methodology as outlined in the 'protocol' described by
Kulmala et al. (2012); Dada et al. (2020). Particle formation rates, at the detection limit of the instrument (typically in the range
2-3 nm), are usually estimated from the time evolution of either the total number concentration or the number concentration
below a certain size (for example 20 nm), correcting for coagulation loss and condensation growth with very simple and
approximate balance equations. For condensational growth, there have been three main approaches: 1) Fitting the growing
nucleation mode with a lognormal function and the growth of the mode GR is defined as the growth of the geometric mean size
of the mode Leppä et al. (2011), 2) The so-called 'maximum concentration method' in which GR is estimated from the times
when each measurement channel of the instrument reaches its maximum concentration Lehtinen and Kulmala (2002), and 3)
The so-called 'appearance time method' in which GR is estimated from concentration rise times of each channel Lehtipalo
et al. (2014). Methods 1 and 2 are applicable to cases, in which there is a clear nucleation mode growing, as is the case, for
example, for the multitude of events analyzed from the Hyytiälä forest station in Finland Maso et al. (2005). For chamber
experiments, where the aerosol size distribution approaches steady state Dada et al. (2020), these two approaches cannot be
used. Method 3 can then be applied to the transition stage of the dynamics, before steady-state is reached. All of these methods
suffer from disturbances by other aerosol microphysical processes (for example coagulation and deposition) and, in addition,
cannot be used to estimate uncertainties related to GR. Deposition rates are typically estimated by targeted experiments with
either several different experiments with different monodisperse aerosol, or in the absence of vapors and with low enough
concentrations that other microphysical processes do not affect the estimation.

The last decade has been a huge leap forward in atmospheric new particle formation research. Instrument development,
especially advances in particle detection efficiency and mass spectrometry, have allowed us to measure details of the dynamics
of the smallest clusters (for example Almeida et al. (2013)). At the same time, however, potentially superior advanced data
analysis methods have not been used. Instead, NPF and particle growth rates have been analyzed with the above mentioned
very simple regression or balance equation approaches (overview of typical methods in Kulmala et al. (2012)), suffering from
potentially crude approximations and permitting no proper estimation of the uncertainties. It is likely that there are significant
inaccuracies in quantities such as particle formation and growth rates estimated previously, and at least their uncertainties have
typically not been quantified. Very recent studies by Kuerten et al. Kürten et al. (2018) have already shown that the difference
between nucleation rates estimated by fitting a sophisticated aerosol model to data and a 'traditional' simple method can be as
large as a factor of ten.





Studies on applying computational inversion methods to estimating the most important quantities of interest with respect to particle fate and effects in the atmosphere, the formation and growth rates, are rare. Lehtinen et al. Lehtinen et al. (2004) applied simple least squares based optimization of aerosol microphysics to measured data. This method was later improved (more processes, less assumptions) by Verheggen et al. Verheggen and Mozurkewich (2006) and Kuang et al. (2012) Kuang et al. (2012). These studies, however, did not address the uncertainties in the estimated parameters. Henze et al. (2004) Henze et al. (2004) used the method of adjoint equations to estimate condensation rates based on measured evolution of the aerosol size distribution. Sandu et al. Sandu et al. (2005) presented adjoint equations of the complete aerosol GDE which could be a basis of data-assimilation of aerosol dynamics. We are, however, not aware of the methodology being used later.

In the *statistical* (Bayesian) framework of inverse problems Kaipio and Somersalo (2006), the uncertainties of the model quantities are modeled statistically, and it offers an approach to uncertainty quantification, in addition to parameter estimation. In time-invariant case, the Bayesian approach was adopted to estimation of aerosol size distributions by Voutilainen et al. Voutilainen et al. (2001). Thus far, the only works where statistical approach has been taken to inverse problems in aerosol size distribution dynamics are those on parameter estimation in aggregation-fragmentation models Ramachandran and Barton (2010); Bortz et al. (2015); Shcherbacheva et al. (2020), estimating the size distribution evolution using Kalman filtering Voutilainen and Kaipio (2002); Viskari et al. (2012) and estimating evaporation rates using a Markov Chain Monte-Carlo method Kupiainen-Määttä (2016). However, the statical inversion framework, and Bayesian state estimation in particular, has been applied to several other problems which are mathematically similar to parameter estimation in aerosol dynamics. Unknown coefficients have been estimated in, for example, Fokker-Planck equations Banks et al. (1993); Dimitriu (2002), age-structured population dynamics models Rundell (1993); Cho and Kwon (1997) as well as algal and phytoplankton aggregation models Ackleh (1997); Ackleh and Miller (2018).

In this paper, we approach the problem of estimating unknown rate parameters in the aerosol general dynamic equation in the framework of Bayesian state estimation. We model the discretized particle size distribution as well as the unknown nucleation, growth and deposition rates in GDE as multivariate random processes, and estimate them from sequential particle counter measurements by using Extended Kalman filter (EKF) and Fixed Interval Kalman Smoother (FIKS). The feasibility of these estimators to quantify the process rate parameters and their uncertainties is tested with series of numerical simulation studies.

## 2 Estimation of parameters in GDE

The temporal evolution of the aerosol size distribution $n = n(d_{\mathrm{p}}, t)$ can be described by a population balance equation referred to as the general dynamic equation (GDE) Zhang et al. (1999); Prakash et al. (2003); Lehtinen and Zachariah (2001);





Smoluchowski (1916); Friedlander and Wang (1966). We write the continuous form of GDE as

$$\frac{\partial n}{\partial t}(d_\mathrm{p},t) = \underbrace{-\frac{\partial g(d_\mathrm{p},t)n(d_\mathrm{p},t)}{\partial d_\mathrm{p}}}_{\text{growth by condensation}} \underbrace{- n(d_\mathrm{p},t)\int\limits_{d_0}^{\infty}\beta(d_\mathrm{p},s)n(s,t)\mathrm{d}s}_{\text{coagulation sink}}$$

$$\underbrace{+\frac{1}{2}\int\limits_{0}^{d_\mathrm{p}}\beta\left(\sqrt[3]{d_\mathrm{p}^3-q^3},q\right)n\left(\sqrt[3]{d_\mathrm{p}^3-q^3},t\right)n(q,t)\mathrm{d}q}_{\text{coagulation source}} \underbrace{-\lambda(d_\mathrm{p},t)n(d_\mathrm{p},t)}_{\text{loss by deposition}} \qquad (1)$$

where $d_\mathrm{p}$ is the particle diameter and $t$ is time. Further, $g = g(d_\mathrm{p},t)$ denotes the condensational growth rate, $\beta = \beta(s, d_\mathrm{p} - s)$ is the coagulation frequency, and $\lambda = \lambda(d_\mathrm{p}, t)$ is the deposition rate.

The boundary conditions consist of fluxes of particles in and out of the considered size range. In reality, the formation of particles occurs at very low size (typically at 1.5 - 2 nm scale) by nucleation; the theoretical size at which molecule clusters

start being stable is referred to as critical size. However, since the measurable size range for size distribution measurements is usually above the critical size, the particle flux to the smallest size class in the considered particle size range, $d_\mathrm{p}^\mathrm{min}$, is driven by condensational growth. Hence, the nucleation $J = J(t)$ is identified with the flux of particles to the smallest size class by condensation, that is,

$$g(d_\mathrm{p}^\mathrm{min},t)n(d_\mathrm{p}^\mathrm{min},t) = J(t). \qquad (2)$$

Similarly, we write for the outward flux of particles at the largest size class

$$g(d_\mathrm{p}^\infty,t)n(d_\mathrm{p}^\infty,t) = 0, \qquad (3)$$

which states that the growth of particles to size exceeding $d_\mathrm{p}^\infty$, the upper limit of the size range, is negligible.

The time- and/or size-dependent parameters $g$, $\beta$, $\lambda$ and $J$ characterize the microphysical properties of aerosols: these parameters, together with boundary conditions of GDE determine completely the evolution of the aerosol size distribution.

However, the process rate parameters are usually not known. In this paper, we aim at estimating the growth, deposition and nucleation rates ($g$, $\lambda$ and $J$, respectively) based on particle size distribution measurements. For the coagulation coefficient $\beta$, a fixed (known/approximate) value will be used.

The analysis proposed in this paper is applicable to both differential mobility particle sizer (DMPS) and scanning mobility particle sizer (SMPS) measurements. For the rest of the paper, however, we refer to measurement modality as SMPS, because

in the numerical example cases, the number of particle size classes is relatively high.

An SMPS measurement is vector $y^k \in \mathbb{R}^M$ which represents an indirect observation of the particle size density $n(d_\mathrm{p}, t_k)$, corrupted by Poisson distributed noise, that is,

$$y^k = \frac{\tilde{y}^k}{V}, \ \text{s.t.} \ \tilde{y}^k \sim \mathrm{Poisson}(Vz^k), \ \text{and} \ z^k = \mathcal{H}n(d_\mathrm{p}, t_k) \qquad (4)$$

where $V$ is a constant (the effective volume of the sample in the condensation particle counter), $\mathcal{H}$ is a device-dependent linear

operator, and $M$ is the number of channels in the particle counter. In the following, we assume that the rate of time evolution





is negligible compared to the time required to measure $M$ channels, or one frame, with SMPS. We denote the time of the measurement of $k^{\text{th}}$ frame by $t_k$.

Since SMPS measurements depend explicitly only on the size density $n$ and not on $g$, $\lambda$ and $J$, a measurement $y^k$ corresponding to a single time instant does not carry enough information for estimating these parameters. However, as $g$, $\lambda$ and $J$

determine the temporal evolution of $n$, it might be possible to estimate them on the basis of a *sequence* of measurements $y^k$ corresponding to a set of time instants $t_k$, $k = 1, \ldots, K$.

In this section, we formulate the problem of estimating the time-dependent size density $n$ and the process rate parameters $g$, $\lambda$ and $J$ as a Bayesian state estimation problem. To this end, we first discretize the GDE with respect to size and time, and write it in a stochastic form in order to model its uncertainties. We also model the process rate parameters as discrete-time stochastic

processes. This formulation allows us to express the following questions in the Bayesian framework Gelb (1974):

- What are the expected values of $n(d_{\text{p}}, t_k), g(d_{\text{p}}, t_k), \lambda(d_{\text{p}}, t_k)$ and $J(t_k)$ at each time $t_k$ given a set of measurements $\mathcal{Y}^\ell = \{y^1, \ldots, y^\ell\}$ corresponding to discrete times $t_1, \ldots, t_\ell$?

- How large are the uncertainties of the estimated quantities?

The state estimation problems are referred to as *prediction, filtering* and *smoothing* depending on whether $\ell < k$, $\ell = k$ or $\ell > k$,

respectively. While filtering is a suitable choice for on-line monitoring and control problems, smoothing is usually a preferable choice when estimates are not needed on-line; the smoother estimates utilize also the future observations $y^{k+1}, \ldots, y^\ell$ for the estimate corresponding to time $t_k$.

The latter question refers to *posterior* uncertainties, that is, uncertainties of the quantities given the measurements $\mathcal{Y}^\ell$. In the Bayesian framework, these uncertainties can be quantified by computing, for example, posterior variances and credible

intervals of the parameters.

In the simplest special case, where the evolution model and observation model are linear with respect to all parameters, and all error terms are additive and Gaussian, the Bayesian filtering and smoothing problems can be solved by Kalman filter and Kalman smoother recursions, respectively Kalman (1960); Gelb (1974). In general cases, where models are non-linear or non-Gaussian, only approximate solutions are available. In principle, the best approximations of the posterior estimates are

obtained with sequential Monte Carlo methods, known as particle filters/smoothers Särkkä (2013). However, these methods are limited to small dimensional cases, because of the high computational burden. Computationally more efficient approximations include ensemble, unscented and extended Kalman filters/smoothers Särkkä (2013). In this paper, we choose the Extended Kalman filter (EKF) and Fixed Interval Kalman Smoother (FIKS), but we note that the other filters and smoothers developed for non-linear state estimation are applicable as well. In the next section, the feasibility of the EKF and FIKS for the GDE

parameter estimation problem will be tested numerically.





## 2.1 Evolution model

### 2.1.1 Discretized, stochastic GDE

To approximate the GDE (1) numerically, we partition the particle size variable $d_p$ into $Q$ intervals (or bins) $\Omega_i$ of widths $\Delta d_i,\ i = 1,\ldots,Q$. The discrete instants of time in the temporal discretization are denoted by $t_k$, $k = 1,\ldots,T$, and the differences between consecutive times by $\Delta t^k = t_{k+1} - t_k$. We denote the number concentration of particles corresponding to $i^{\text{th}}$ bin at time $t_k$ by $N_i^k$, that is, $N_i^k = \int_{\Omega_i} n(d_p,t_k)\mathrm{d}d_p$, and a vector consisting of particle concentrations in all $Q$ bins at time $t_k$ by $N^k$, that is, $N^k = [N_1^k,\ldots,N_Q^k]^{\mathrm{T}}$. We discretize the condensation and deposition rates accordingly, and write $g^k = [g_1^k,\ldots,g_Q^k]^{\mathrm{T}}$, $\lambda^k = [\lambda_1^k,\ldots,\lambda_Q^k]^{\mathrm{T}}$. Further, the nucleation $J$ is discretized with respect to time: $J^k$ denotes the nucleation rate at time $t_k$.

Using Euler's method for time integration and 1st order upwinding differencing for the condensation terms, we get the discrete-time evolution model for the particle number concentrations in bins $\Omega_i$ Korhonen et al. (2004):

$$N_1^{k+1} = N_1^k + \Delta t^k \left( J^k - \left( \tfrac{g_1^k}{\Delta d_1} + \lambda_1 \right) N_1^k \right) + C_1(N^k) \tag{5}$$

$$N_i^{k+1} = N_i^k + \Delta t^k \left( \tfrac{g_{i-1}^k}{\Delta d_{i-1}} N_{i-1}^k - \left( \tfrac{g_i^k}{\Delta d_i} + \lambda_i \right) N_i^k \right) + C_i(N^k), \tag{6}$$
$$\text{for all } 1 < i \le Q,\ 1 \le k \le T$$

where $C_i(N^k)$ is a non-linear coagulation term; for details, see Lehtinen and Zachariah (2001). Equations (5) – (6) can be written equivalently in vector form as

$$N^{k+1} = A(g^k,\lambda^k)N^k + s(J^k) + C(\beta,N^k) \tag{7}$$

where $A = A(g^k,\lambda^k)$ is a sparse matrix consisting of elements $A_{ij}$:

$$A_{ij}(g^k,\lambda^k) = \begin{cases} 1 - \Delta t^k \left( \tfrac{g_i^k}{\Delta d_i} + \lambda_i \right) & i = j \\ \Delta t^k \tfrac{g_{i-1}^k}{\Delta d_{i-1}} & i = j+1 > 1 \\ 0 & \text{otherwise,} \end{cases} \tag{8}$$

$s = s(J^k) \in \mathbb{R}^Q$ is a vector of the form $s(J^k) = [\Delta t^k J^k, 0, \ldots, 0]^{\mathrm{T}}$, and the non-linear term $C(\beta,N^k)$ is defined accordingly.

Finally, we complement the discretized GDE with a stochastic term $\epsilon_k \in \mathbb{R}^Q$ to account for modeling errors caused by, for example, discretization and uncertainties of the boundary conditions, and write

$$N^{k+1} = f(N^k) + \epsilon_k \tag{9}$$

where $f : \mathbb{R}^Q \to \mathbb{R}^Q$ is of the form $f(N^k) = A(g^k,\lambda^k)N^k + s(J^k) + C(\beta,N^k)$. In this paper, the stochastic state noise term $\epsilon_k$ is modeled as Gaussian, $\epsilon_k \sim \mathcal{N}(0,\Gamma_\epsilon)$, where $\Gamma_\epsilon$ is the covariance matrix of $\epsilon_k$. Equation (9) forms a discretized, stochastic evolution model for the particle number concentration $N$.





### 2.1.2 Models for the parameters of GDE

All the unknown parameters of GDE ($g$, $\lambda$ and $J$) are known to be non-negative. For this reason, we reparametrize these quantities by writing $g_i^k = P_g(\xi_{g,i}^k)$, $\lambda_i^k = P_\lambda(\xi_{\lambda,i}^k)$ and $J^k = P_J(\xi_J^k)$, where $\xi_{g,i}$, $\xi_{\lambda,i}$ and $\xi_J$ are the (unconstrained) parameters,

and mappings $P_g$, $P_\lambda$, $P_J$ have the form of a so-called softplus function Dugas et al. (2001) $P_\varphi : \mathbb{R} \to \mathbb{R}^+$

$$P_\varphi(\xi_\varphi^k) = \frac{1}{\alpha} \log \left( 1 + e^{\alpha \xi_\varphi^k} \right). \tag{10}$$

We denote the vectors consisting of all condensation and deposition parameters at time $t_k$ by $\xi_g^k = [\xi_{g,1}^k, \dots, \xi_{g,Q}^k]^\mathrm{T}$ and $\xi_\lambda^k = [\xi_{\lambda,1}^k, \dots, \xi_{\lambda,Q}^k]^\mathrm{T}$, respectively.

As the process rates $g$, $\lambda$ and $J$ in GDE are time-varying, the state estimation requires modeling their time dependence. In

this paper, we model $\xi_{g,i}$, $\xi_{\lambda,i}$ and $\xi_J$ either as first order Markov processes

$$\xi_\varphi^{k+1} = \Psi_\varphi \xi_\varphi^k + \eta_\varphi^k \tag{11}$$

where $\Psi_\varphi$ is a diagonal matrix $\Psi_\varphi = r_\varphi I$, and $r_\varphi \in ]0,1[$, or as second order Markov processes

$$\xi_\varphi^{k+2} = \Psi_\varphi^1 \xi_\varphi^{k+1} + \Psi_\varphi^2 \xi_\varphi^k + \eta_\varphi^k \tag{12}$$

with $\Psi_\varphi^1 = r_\varphi^1 I$, $\Psi_\varphi^2 = r_\varphi^2 I$. In both models, $\eta_\varphi^k$ stands for Gaussian noise $\eta_\varphi^k \sim \mathcal{N}(0, \Gamma_{\eta_\varphi^k})$. To simplify the following descrip-

tion, we assume all the models to be of the form (11), but we note that the extension to second order models is straightforward: Higher order Markov models can be converted into the form of first order Markov models, by augmenting the state variables corresponding to more than one time instant into a single vector. The second order models are suitable for some of the quantities in GDE, because they imply temporal smoothness of those processes Kaipio and Somersalo (2006). The specific choices of the state models and their parameters are discussed in Section 3

The evolution models, such as (11 - 12) can be argued to be unrealistic, as they are not based on physics. The understanding, however, is that if the (co)variances of the driving noise processes $\eta_\varphi^k$ are set high enough, such models are feasible in the sense that the *actual* $\xi_\varphi^{k+1} - \Psi_\varphi \xi_\varphi^k$ are well supported by the modelled distribution of $\eta_\varphi^k$. There are systematic (state-space identification) approaches that allow for testing the feasibility of the model for the driving noise distribution (variances) Gelb (1974). In the next section, we test the state estimation based on the above models in cases, where the true evolution of the

quantities is *not* of the form of Markov models.

### 2.1.3 Augmented evolution model for $n$, $g$, $\lambda$ and $J$

To complete the evolution model, we define an augmented state variable $X^k$,

$$X^k = \begin{bmatrix} N^k \\ \xi_g^k \\ \xi_\lambda^k \\ \xi_J^k \end{bmatrix} \tag{13}$$





and combine the evolution models written in Sections 2.1.1 and 2.1.2, yielding

$$
\begin{bmatrix} N^{k+1} \\ \xi_g^{k+1} \\ \xi_\lambda^{k+1} \\ \xi_J^{k+1} \end{bmatrix} = \begin{bmatrix} A(P_g(\xi_{g,i}^k), P_\lambda(\xi_{\lambda,i}^k)) & 0 & 0 & 0 \\ 0 & \Psi_g & 0 & 0 \\ 0 & 0 & \Psi_\lambda & 0 \\ 0 & 0 & 0 & \Psi_J \end{bmatrix} \begin{bmatrix} N^k \\ \xi_g^k \\ \xi_\lambda^k \\ \xi_J^k \end{bmatrix} + \begin{bmatrix} s(P_J(\xi_J^k)) \\ 0 \\ 0 \\ 0 \end{bmatrix} + \begin{bmatrix} C(\beta, N^k) \\ 0 \\ 0 \\ 0 \end{bmatrix} + \begin{bmatrix} \epsilon^k \\ \eta_g^k \\ \eta_\lambda^k \\ \eta_J^k \end{bmatrix} \tag{14}
$$

or

$$
X^{k+1} = F(X^k) + w^k. \tag{15}
$$

This is the evolution model for the augmented state variable $X^k$, which includes not only the number concentrations of the size sections but also the unknown process rates. Next, we write the observation model in terms of $X^k$, and then, in Section 2.3, we apply Bayesian state estimation to infer $X^k$, $k = 1, \ldots, K$ based on sequential SMPS measurements.

## 2.2 Observation model

A scanning mobility particle sizer (SMPS) consists of a differential mobility analyzer (DMA), which classifies charged particles based on their mobility in an electric field, and a condensation particle counter (CPC) where the classified particles are grown to sizes detectable for example optically. All particle counters provide only discrete, indirect and noisy data on the particle size distributions. Mathematically, each channel in a particle counter gives data that corresponds to convolution / projection of the particle size distribution onto a space spanned by device-specific kernel functions; moreover, the counter data is corrupted by Poisson distributed noise.

### 2.2.1 SMPS transfer function

The output of each DMA channel $i$ corresponding to a discrete time index $k$ is of the form

$$
z_i^k = \frac{1}{V} \int\limits_{t_0+(k-1)\Delta t}^{t_0+k\Delta t} \phi_a(t) \int\limits_{\omega_i} \psi_i(d_{\mathrm{p}}) n(d_{\mathrm{p}}, t) \mathrm{d}d_{\mathrm{p}} \mathrm{d}t \tag{16}
$$

where $\psi_i(d_{\mathrm{p}})$ is a time invariant kernel function, $\omega_i$ is the support of $\psi_i(d_{\mathrm{p}})$, that is, the set of points where $\psi_i(d_{\mathrm{p}})$ is non-zero, $t_0$ is the initial time and $\Delta t$ is the duration of counting particles in the CPC for a single channel. Further, $V$ is the volume of the aerosol sample that passes through the CPC counter with a detector-sample-flow rate $\phi_a(t)$ in the period of time $\Delta t$, that is, $V = \int_{t_0+(k-1)\Delta t}^{t_0+k\Delta t} \phi_a(t) \mathrm{d}t$.





### 2.2.2 CPC counting model

The measurement data of the $i^{\text{th}}$ channel of SMPS, $y_i^k$, consists of Poisson distributed counts given by CPC:

$$y_i^k = \frac{\tilde{y}_i^k}{V}, \quad \text{with} \quad \tilde{y}_i^k \sim \text{Poisson}(V z_i^k) \tag{17}$$

where $V \simeq \Delta t \phi_a(t_0 + k\Delta t)$ is the volume of sample used in the CPC for counting. In the numerical studies of this paper, we use this model when simulating the measurement data. In state estimation, however, we approximate the Poisson distributed observations as Gaussian:

$$y_i^k \sim \mathcal{N}\left(z_i^k, \gamma_i\right) \tag{18}$$

where $\gamma_i$ is the approximate variance of the noise. In this paper, we use the same approximation as in Voutilainen et al. (2000) and write $\gamma_i^k = \frac{y_i^k}{V}$.

By discretizing the SMPS model (16) and using the above Gaussian approximation of the noise, we write an observation model of the form

$$y^k = \bar{H}N^k + \tilde{e}^k + \iota^k, \tag{19}$$

where $y^k = [y_1^k, \ldots, y_M^k]^{\text{T}}$, $\bar{H}$ is an observation matrix and $\tilde{e}^k$ is the Gaussian observation noise $\tilde{e}^k \sim \mathcal{N}(0, \Gamma_{\tilde{e}}^k)$. Here the covariance of the observation noise is of diagonal form $\Gamma_e^k = \text{diag}(\gamma_1^k, \ldots, \gamma_M^k)$. The additional noise term, $\iota_k$ is included in Equation (19) in order to account for errors caused by the discretization of the measurement operator in Equation (16). Here, $\iota_k$ is simply approximated as Gaussian distributed and zero-mean, $\iota^k \sim \mathcal{N}(0, \Gamma_\iota^k)$ and its components as mutually independent; hence, the covariance matrix $\Gamma_\iota^k$ is of diagonal form. Furthermore, the approximation error term $\iota_k$ is assumed to be independent of the counting noise $\tilde{e}^k$, and thus, the total error $e^k = \tilde{e}^k + \iota^k \sim \mathcal{N}(0, \Gamma_e^k)$, where $\Gamma_e^k = \Gamma_{\tilde{e}}^k + \Gamma_\iota^k$. For more rigorous approach to handling modeling errors, we refer to book Kaipio and Somersalo (2006).

Finally, the model (19) can be written in terms of the state variable $X^k$ defined in Equation (13),

$$y^k = HX^k + e^k, \tag{20}$$

where $H$ is a block matrix of the form $H = [\bar{H}, 0, 0, 0]$.

### 2.3 State estimation

The non-linear evolution model (15) and the linear observation model (20) form a system

$$
\begin{aligned}
X^{k+1} &= F(X^k) + w^k \tag{21}\\
y^k &= HX^k + e^k, \tag{22}
\end{aligned}
$$

referred to as the state-space representation. The system is stochastic due to the state noise and observation noise processes, $w^k$ and $e^k$, respectively. In addition, we model the initial state $X^0$ as a Gaussian random variable $X^0 \sim \mathcal{N}(X^{0|0}, \Gamma^{0|0})$. Given





this model, we can state the Bayesian filtering and smoothing problems as: *Form the conditional probability density of the*
*random variable $X_k$, given the sequence of measurements $\mathcal{Y}^\ell = \{y^1, \ldots, y^\ell\}$.* In Extended Kalman filter and smoother, these probability densities $\pi(X^k|\mathcal{Y}^\ell)$, are approximated by Gaussian densities, that is,

$$\pi(X^k|\mathcal{Y}^\ell) \approx \mathcal{N}(X^{k|\ell}, \Gamma^{k|\ell}), \tag{23}$$

where $X^{k|\ell}$ and $\Gamma^{k|\ell}$, respectively, are approximations of the conditional expectation and covariance of $X_k$ given $\mathcal{Y}^\ell$ Gelb (1974).

Kalman filtering gives online estimates $\pi(X^k|\mathcal{Y}^k)$ based on the data set from beginning up to the present time $k$, that is, $\ell = k$, while in smoothing $\ell > k$. In Fixed Interval Kalman Smoother (FIKS), in particular, $\ell = K$, where $K$ is the index of the final time step. In other words, the FIKS estimate $\pi(X^k|\mathcal{Y}^K)$ at each time $k$ is based on the entire data set from the beginning to the end of measurements. The EKF and and FIKS estimates (approximate expectations $X^{k|\ell}$ and covariances $\Gamma^{k|\ell}$) are given by the following forward and backward iterations Gelb (1974):

---

**Algorithm 1** Extended Kalman Filter (EKF)

**Initial state:** Expectation $X^{0|0}$ and covariance $\Gamma^{0|0}$

**for** $k = 1, \ldots, K$ **do**

   Prediction: expectation and covariance
$$X^{k|k-1} = F\left(X^{k-1|k-1}\right)$$
$$\Gamma^{k|k-1} = \partial F^{k-1} \Gamma^{k-1|k-1} (\partial F^{k-1})^{\mathrm{T}} + \Gamma_w^{k-1}$$

   Kalman gain matrix:
$$K^k = \Gamma^{k|k-1} (H^k)^{\mathrm{T}} (H^k \Gamma^{k|k-1} (H^k)^{\mathrm{T}} + \Gamma_e^k)^{-1}$$

   Measurement update: filter expectation and covariance
$$X^{k|k} = X^{k|k-1} + K^k (y^k - H^k X^{k|k-1})$$
$$\Gamma^{k|k} = (I - K^k H^k) \Gamma^{k|k-1}$$

**end**

---


---

**Algorithm 2** Fixed Interval Kalman Smoother (FIKS)

**Initialization:** Run EKF (Algorithm 1), store all variables

**for** $k = K, \ldots, 1$ **do**    Backward gain matrix
$$A^k = \Gamma^{k|k} (\partial F)^{\mathrm{T}} (\Gamma^{k+1|k})^{-1}$$

   Smoother expectation and covariance
$$X^{k|k} = X^{k|K} + A^k \left(X^{k+1|K} - X^{k+1|k}\right)$$
$$\Gamma^{k|K} = \Gamma^{k|k} + A^k \left(\Gamma^{k+1|K} - \Gamma^{k+1|k}\right)(A^k)^{\mathrm{T}}$$

**end**

---





In Algorithms 1 and 2, $\partial F^k$ denotes the Jacobian matrix of the non-linear mapping $F\left(X^k\right)$ at point $X^{k|k}$. Note that the FIKS is based on a backward recursion, which starts from the filter estimate corresponding to final state: $X^{K|K}, \Gamma^{K|K}$.

## 3   Numerical simulations

In this section, the feasibility of the proposed estimation scheme is tested with numerical studies, where aerosol particle
evolutions are simulated by numerical approximations of the GDE corresponding to a set of process rates, and where synthetic SMPS data is computed by numerical modeling of the DMA and generating the Poisson distributed CPC data. Two type of events are considered: In Cases 1 and 2, the evolution of the aerosol size distribution is governed by a nucleation event (NE) and the subsequent growth of the nucleation mode in a background of existing aerosols. In Cases 3 and 4, new particles are formed in a continuous nucleation process and their further evolution is controlled by condensational growth and deposition,
so that the size distribution approaches a steady state (SS). In all cases, the particle growth is dominated by condensation, and the loss of particles (caused by wall deposition and sedimentation) depends linearly on the size distribution function. Further, the coagulation kernel is chosen to have a form given in book Seinfeld and Pandis (2016), for details see Appendix A. In these numerical studies, Cases 1 and 2 (NE) represent qualitatively a typical particle formation event in the atmosphere (for example Hyytiälä; Dal Maso et al., 2005 Maso et al. (2005)), whereas Cases 3 and 4 (SS) represent particle formation and growth in a
chamber experiment (for example CLOUD; Lehtipalo et al., 2014 Lehtipalo et al. (2014)).

### 3.1   Cases 1 & 2: Nucleation event (NE)

#### 3.1.1   NE: Data simulation

In the numerical simulation study, the temporally evolving particle size distribution is synthetically generated by using the (deterministic) discretized GDE model described by Equations (5) and (6) with predefined process rates $g, \lambda, \beta$ and $J$. In the
estimation, however, $g, \lambda$ and $J$ are, of course, not known.

In the NE case, the process rates $g, \lambda$ and $J$ are chosen to have the following properties: The condensational growth rate $g$ is independent of particle size but depends on time, while the deposition rate $\lambda$ depends on particle size but not on time. Further, the nucleation rate $J$ is a time-dependent, continuous function which represents an NE in a time interval $[t_0 = 5\text{h}, t_1 = 10\text{h}]$ and is zero in all other instants within the period of interest $[0, 15\text{h}]$. As noted above, the coagulation kernel is known, yet the
coarse discretization causes error also to this term. The detailed forms of the process rates are shown in Table A1 (Appendix A).

When simulating the evolution of the particle distribution, the particle diameter $d_p \in [13.85\,\text{nm}, 1000.0\,\text{nm}]$ is discretized into $Q = 2500$ logarithmically distributed bins. Such a high size resolution is chosen in order to avoid numerical diffusion effects and to obtain a good approximation of the particle size density. The time step $\Delta t^k$ in the explicit Euler time integration



scheme is chosen based on the Courant–Friedrichs–Lewy condition Courant et al. (1928); Dullemond and Dominik (2005):

$$0 < \Delta_t^k < \frac{1}{\max\limits_i \{ \frac{g_i^k}{\Delta d_i} + \lambda_i \}}. \tag{24}$$

The CFL criterion is applied in order to keep the time-integration stable with respect to condensational growth and deposition. Coagulation rates are not considered here, because coagulation is actually a dampening mechanism that stabilizes time integration. The nucleation, growth and deposition rates as well as the resulting particle size density evolution for the NE cases are illustrated in Figure 1.

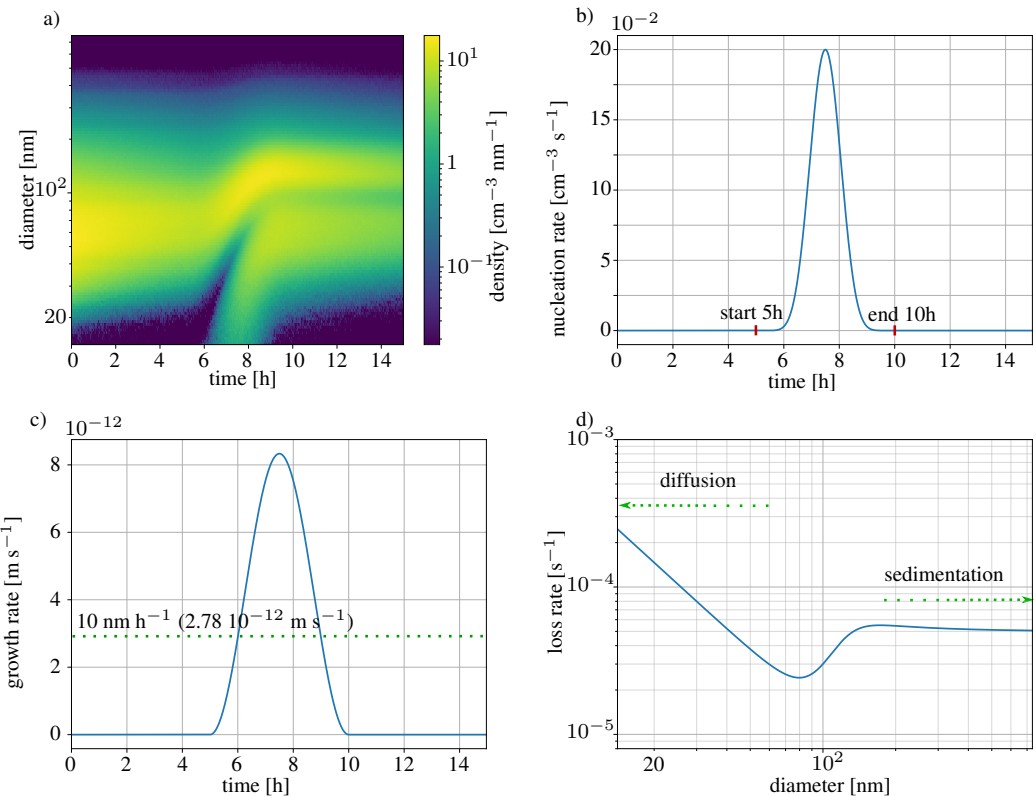

**Figure 1.** Cases 1 & 2, nucleation event; true (simulated) processes: a) particle size density (measurement), b) nucleation rate, c) growth rate, and d) wall loss rate.


As explained in Section 2.2, the measurements consist of simulated counts modeling CPC combined with DMA. The model for the kernel functions $\psi_i(d_p)$ corresponds to an SMPS3936 device, and it accounts for the most relevant effects Millikan (1923); Stolzenburg (1989); Flagan (1998); McMurry (2000); Boisdron and Brock (1970); Wiedensohler (1988). We skip the details here, and only visualize the kernels, by plotting the size distribution transfer function, or the observation matrix $\bar{H}$, as

a colormap (Figure 2 a)).




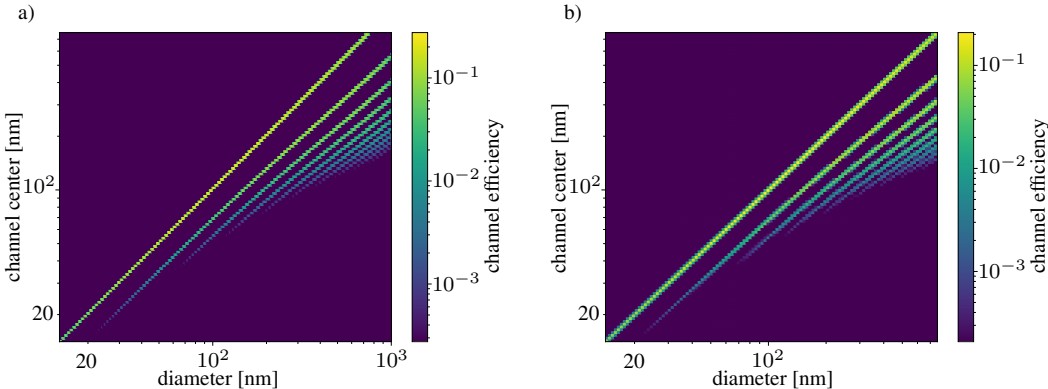

**Figure 2.** The size distribution transfer functions for a) simulating the measurement data, and b) observation model used in state estimation. In the images, the horizontal axis corresponds to the size of the particle entering the device, and the vertical axis represents the channels of the SMPS. The colors represent the values of the efficiency with which a given particle size will be classified in a given bin in particle sizer.

We simulate the synthetic CPC measurement $y_i^k$ corresponding to each CPC channel $i$ at each time $k$ by drawing samples from a Poisson distribution, given in Equation (17) with mean $Vz_i^k$. Since the expectation of $y_i^k$ is $z_i^k$ and its variance $z_i^k/V$, the signal-to-noise ratio (SNR) of CPC data increases with $V$. In order to investigate the effect of SNR to the process rate estimates, we generate the Poisson-distributed observations corresponding to two sample volumes: $V = 90\,\mathrm{cm}^3$ (Case 1: NE, high SNR) and $V = 0.9\,\mathrm{cm}^3$ (Case 2: NE, low SNR).

### 3.1.2 NE: Parameter estimation

In this section, we briefly describe the assumptions made when constructing the models used for computing the state estimates in the NE cases. The exact forms of the models as well as choices of parameters are listed in Appendix B.

For evolution models used in state estimation, the particle size range $[14.1\,\mathrm{nm}, 736.5\,\mathrm{nm}]$ is divided into 111 bins, that is, the size range is narrower and the discretization is significantly coarser than when simulating the data. The GDE-based, discrete stochastic state evolution model (9) for the particle number $N$ is written as described in Section 2.1.1. The covariance matrix $\Gamma_\epsilon$ of the stochastic term $\epsilon_k$ is chosen to be of diagonal form.

In this case study, we assume to know that the condensation rate is time-dependent but size-independent. Moreover, both the condensation and nucleation rates are assumed to be temporally smooth, and we model them as second order stochastic processes (Equation (12)). The wall loss factor $\lambda$ is assumed to be a smooth function of the particle size. Further, we write a first order Markov model (Equation (11)) for its temporal variation, that is, although the true wall loss factor is time-invariant, we do not assume to know this property in state estimation. This is done to study the stability of the estimation scheme: although the wall loss is modeled as time-varying, the estimation should yield essentially time-invariant estimates.

The covariance of the initial state, $\Gamma_{0|0}$ is chosen to be diagonal; this signifies that the elements of $X_0$ are mutually independent. Moreover, the variances of $X_0$ are chosen to be relatively large in comparison with the variances of the state noise vector





$\epsilon_k$; this indicates a high uncertainty of the initial state. We note that the selection of the parameters in the stochastic terms is a crucial part of the state-space model. However, the state estimates are not extremely sensitive to these choices; choosing parameters that are of right order of magnitude is usually enough – and since the stochastic models are written for physically relevant quantities, ballpark *ranges* of the parameters are often available *a priori*.

The size distribution transfer function corresponding to the discretization of the particle size in estimation is illustrated in Figure 2 b). The approximate observation model is used in order to avoid so-called *inverse crime*, which means the use of unrealistically accurate models in the inversion of simulated data. Secondly, as noted in Section 2.2, instead of using the Poisson model for the measurements, the observation noise is approximated as additive and Gaussian. This choice is made for computational convenience, as it allows for the direct applications of EKF and FIKS into the state-space system.

The Extended Kalman filter and smoother estimates are computed using Algorithms 1 and 2, respectively. From the resulting state estimates $X^{k|\ell}$, $\ell = k, K$ the approximate posterior expectations of the processes are computed using the models $g_i^k = P_g(\xi_{g,i}^k)$, $\lambda_i^k = P_\lambda(\xi_{\lambda,i}^k)$ and $J^k = P_J(\xi_J^k)$ (see Sections 2.1.2 and 2.1.3). We also compute approximate 68 % credible intervals of the estimates, by mapping the values $E(\xi_{*,i}^k|\mathcal{Y}^\ell) \pm \sqrt{\mathrm{var}(\xi_{*,i}^k|\mathcal{Y}^\ell)}$, to the corresponding process rate spaces. Note, however, that due to the linearizations / Gaussian approximation behind EKF, these approximate intervals do not necessarily represent

the ranges within which the true parameter value is with probability 68 %. In the following, we refer to these approximate credible intervals as *posterior error intervals*.

### 3.1.3   NE: Results and discussion

The results of Case 1 (NE, high SNR) are illustrated in Figure 3. The figure shows the Kalman smoother estimates for the particle size density, and both the Kalman filter (EKF) and smoother (FIKS) estimates for the growth, nucleation and wall loss

rates as well as for two instantaneous particle size densities ($\frac{\Delta N}{\Delta d_p}$ at 2h and 10h). The wall loss rate estimates correspond to time 10h. For the process rates and for the instantaneous size densities, the figure also shows the EKF and FIKS based posterior error intervals – representing the uncertainty of the estimates – as well as the true values of the corresponding quantities.

    The estimated particle size density (Figure 3 a)) is in rather good correspondence with the true size density (Figure 1 a)). However, the size density estimates corresponding to times 2h and 10h (Figure 3 e)-f)) show that the peak values of the size

density are somewhat underestimated by both EKF and FIKS. Moreover, in these instants, the true values of the size density are partly outside the approximate posterior error intervals. Yet the 68 % posterior error intervals do not necessarily need to contain the true values, the error intervals seem to be slightly too narrow. This underestimation of the uncertainty is due to the linearizations/Gaussian approximation behind EKF and FIKS, as shown in Huttunen et al. (2018). The errors caused by model approximations become more influential with decreasing mean noise level – the remedy for such errors would be the Bayesian

approximation error method Kaipio and Somersalo (2006), which, however, is out of the scope of this paper.

    For the process rates, the approximate posterior means given by both EKF and FIKS are relatively close to true values (Figure 3 b)–d)). Overall, the FIKS estimates for the process rate parameters are more accurate than the EKF estimates – this is an expected result, because FIKS utilizes the entire data set up to the end of the process, while EKF uses only data up to time $t$ when estimating the variables at time $t$.







**Figure 3.** Case 1: NE, high SNR. State estimates for the particle size density (subfigures a, e and f), nucleation rate (b), growth rate (c), and wall loss rate (d). The image in plot a) depicts the approximate posterior expectation for the entire time-evolution of the particle size density given by FIKS, while e) and f) illustrate the EKF and FIKS estimates corresponding to times 2h and 10h, respectively. In plots b)–f), the blue and orange lines represent the approximate posterior expectations for EKF and FIKS, respectively, and the areas shaded with light blue and orange are the respective posterior error intervals. The true values of the corresponding quantities are drawn with green line.





The filter and smoother show also differences in the posterior error intervals of the process rate parameters: The error intervals given by EKF are systematically wider than those given by FIKS. This is again an expected result, because the use of the future data (FIKS) should, indeed, reduce the uncertainty in the estimated quantities. Furthermore, in almost all instants of time, the process rate parameters (especially growth and nucleation rates) are within the posterior error intervals. This is a desired result, as it indicates that these approximate credible intervals give realistic measures of the estimate uncertainties in

these cases.

    The wall loss rate estimate uncertainty depends strongly on the particle size: The posterior error intervals are wide in the lowest and highest size ranges, and rather narrow elsewhere. The high uncertainty of the wall loss factor at the high particle size range $d_p > 400$ nm is caused by the lack of data – in this size range, the particle density is nearly zero at all times, and consequently, the SMPS data does not provide information on the wall loss factor. In the lowest size range, the width of the

credible interval depends also on time. From time $t = 9$ h, when the nucleation event is over, the particle size density in the lowest size range is almost zero, resulting again in high uncertainty in the wall loss factor estimate.

    In Case 1, the SNR is high, and – apart from the aforementioned exceptions – the estimate uncertainty is very low. Figure 4 shows the results of Case 2, where the SNR is significantly lower. As exptected, the process rate estimates become less accurate than in Case 1. However, the change in the accuracy is quite small, especially for FIKS – demonstrating that the

Kalman smoother estimates tolerate measurement noise rather well. As expected, the posterior error intervals of all estimated quantities are clearly wider than in the case of high SNR – this is a result of increased uncertainty in the particle counter observations. Both EKF and FIKS lead to safe posterior error intervals for the process rate parameters, and again, FIKS gives clearly more narrow posterior error intervals than EKF. Again, the true values of the process rate parameters are within the posterior error intervals, further confirming the feasibility of Bayesian state estimation to quantifying the uncertainties of the

process rates.

### 3.2   Cases 3 & 4: Steady state (SS)

#### 3.2.1   SS: Data simulation and parameter estimation

In the SS simulations of this paper, the condensation rate is both size- and time-dependent, and the wall loss rate is time-invariant but depends on size. Both the nucleation and condensation rates start from zero and grow within the first 30 min until

they reach their stationary values. In such a case, the new particle formation and condensational growth are compensated by wall losses, leading to the number concentration function to reach a steady state. The simulated process rates and the particle size density are illustrated in Figure 5. Here, the size range of particles is $[0.87\,\text{nm}, 10.00\,\text{nm}]$, and it is discretized into $Q = 1731$ logarithmically distributed bins.

    The synthetic SMPS data is simulated similarly to Cases 1 and 2. Again, two cases corresponding to different SNRs are

simulated, by generating the Poisson-distributed observations corresponding to two sample volumes: $V = 200$ cm$^3$ (Case 3: SS, high SNR) and $V = 2$ cm$^3$ (Case 4: SS, low SNR).





**Figure 4.** Case 2: NE, low SNR. State estimates for the particle size density (subfigures a, e and f), nucleation rate (b), growth rate (c), and wall loss rate (d). The image in plot a) depicts the approximate posterior expectation for the entire time-evolution of the particle size density given by FIKS, while e) and f) illustrate the EKF and FIKS estimates corresponding to times 2 h and 10 h, respectively. In plots b)–f), the blue and orange lines represent the approximate posterior expectations for EKF and FIKS, respectively, and the areas shaded with light blue and orange are the respective posterior error intervals. The true values of the corresponding quantities are drawn with green line.



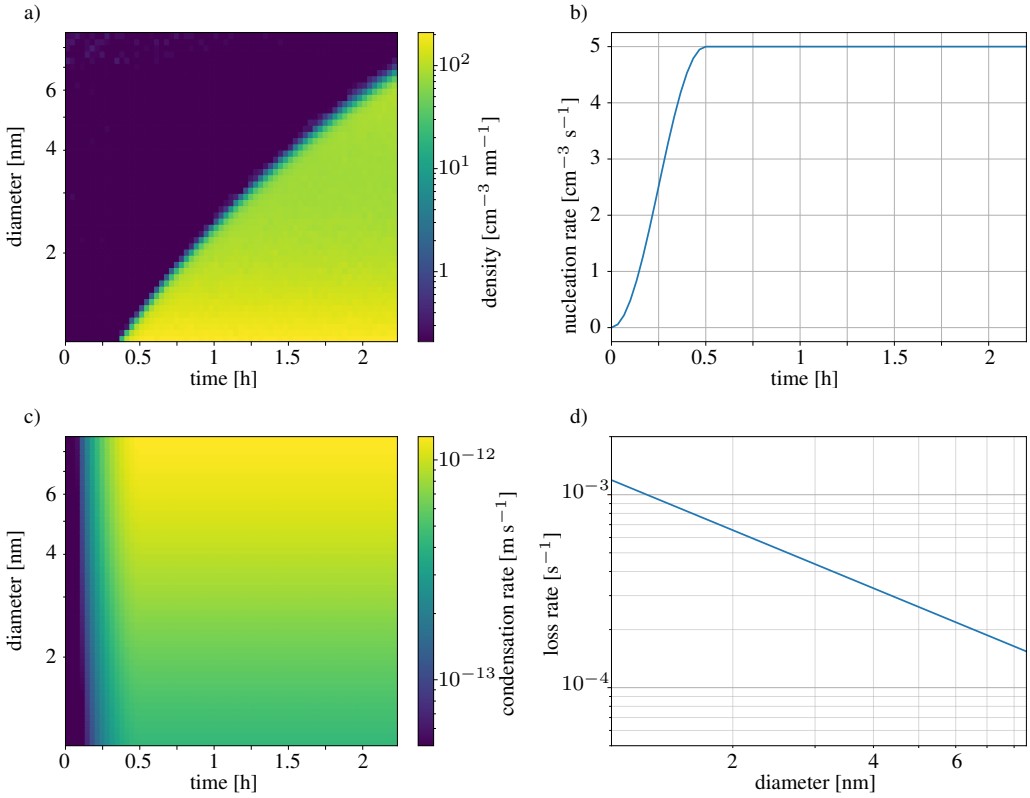

**Figure 5.** Cases 3 & 4, steady state system; true (simulated) processes: a) particle size density (measurement), b) nucleation rate, c) growth rate, and d) wall loss rate.

The time evolutions of the nucleation, growth and wall loss rates are modeled as first order Markov processes (Equation (12)). The growth and wall loss rates are assumed to be smooth with respect to size, and hence, the state noises in the corresponding evolution models are modeled as correlated. For details of the model choices we refer to Appendix 2.

### 390 3.2.2 SS: Results and discussion

Figure 6 illustrates the estimates of the particle size density and process rates in the case of high SNR (Case 3). For the particle size density and condensation rate, the images in Figs. 6 a) and c), respectively, show the approximate FIKS-based posterior means of those time- and size-dependent variables. For the nucleation rate as well as for the instantaneous wall loss rate and size densities, both the filter and smoother estimates (approximate posterior expectations and error intervals) are plotted.

In this case, the smoother estimate for the particle size density is in very good correspondence with the true density (cf. Figure 5 a)). Further, the uncertainty estimates for the particle size density are feasibile: The true size density lays within the posterior error intervals given by EKF and FIKS.





The nucleation rate is clearly underestimated by both EKF and FIKS. The EKF based posterior error intervals are wide, indicating high uncertainty in the estimate, and the true nucleation rate lays mostly inside the EKF posterior error interval. The

FIKS-based posterior error interval, on the other hand, is too narrow as the true nucleation rate is outside it. This is again a consequence of the modelling errors discussed in Section 3.1.3: While the true nucleation rate plotted in green takes place at 0.87 nm, in the state-space model, the diameter that corresponds to nucleation is about 1.08 nm (the lower end of the smallest size bin, geometrical center at 1.1 nm). This difference between the model used for simulating the data and the model used in estimation causes the systematic error — underestimation of the nucleation rate. Further, because this modelling error is not

accounted for in estimation, the approximate credible intervals of the nucleation rate are too narrow.

Figure 6 c) shows a clear trend in the quality of the smoother estimate for the condensational growth rate: At an early stage of the process (time $\sim 0.25$ h), FIKS infers the growth rate reliably only in the smallest size classes (diameter $\sim 1$ nm); in the larger particle sizes, the growth rate is heavily underestimated (cf. Figure 5 c)). As time progresses, the growth rate estimates become gradually more reliable in larger and larger size classes.

The gradual improvement in the growth rate estimates in the larger size classes is a direct consequence of the propagation of the particle number density towards large size classes. Indeed, comparison of Figures 6 a) and c) reveals that the size class where FIKS catches the increase in the growth rate parameter (light/yellow area in the condensation rate image Figure 6 c)) follows accurately the propagating front of the number density in Figure 6 a). The reason for this property of the growth rate estimate is obvious: In the size classes where the particle number density is very low, the measurement data does not

carry information on the growth rate parameters. In the beginning of the process, the particle number density is low in all classes. When nucleation starts producing particles to the smallest size class and these particles grow, the growth is sensed by the particle size analyzer measurements. When the particle sizes keep increasing due to condensation, the measurements corresponding to increasingly larger size classes become sensitive to the process rate parameters.

Figure 7 illustrates the EKF and FIKS estimates of the growth rate corresponding to four instants of time. These plots

confirm the above discussion on the growth rate estimation: At time 45 min, both the EKF and FIKS based posterior means underestimate the growth rate in the large size classes (above $\sim 1.5$ nm), and in the subsequent times, the EKF and especially FIKS estimates become reliable in gradually increasing size classes. Furthermore, Figure 7 shows a trend in the evolution of the posterior error intervals of the growth rate: At time 45 min, the posterior error intervals are really wide in classes $> 1.5$ nm, reflecting high uncertainty in the growth rate estimates in this size range. As time progresses, the size range of low uncertainty

spreads towards large size classes. This result demonstrates that Bayesian filtering and smoothing yield feasible posterior error estimates – indicating high uncertainty in the size ranges where the posterior expectations are unreliable in this example.

The EKF estimates of the growth rate in Figure 7 show similar behaviour as the FIKS estimates; the main differences are that, as expected, the posterior means given by EKF are more biased than those in FIKS, and posterior error intervals are overall wider than with FIKS.

Figures 8 and 9 show the results of Case 4: SS and low SNR data. The properties of the state estimates are very similar to those in Case 3, except for the anticipated differences: The lower accuracy of the approximate posterior means, and wider posterior error intervals – again, indicating the increase of the uncertainty when the SNR of the measurements gets lower.





**Figure 6.** Case 3: SS, high SNR. State estimates for the particle size density (subfigures a, e and f), nucleation rate (b), growth rate (c), and wall loss rate (d). The images in plots a) and c) depict the approximate, FIKS-based posterior expectations for the entire time-evolutions of the corresponding quantities. Plots e) and f) illustrate the EKF and FIKS estimates for the particle size density corresponding to times 2 h and 10 h, respectively. In plots b) and d)–f), the blue and orange lines represent the approximate posterior expectations for EKF and FIKS, respectively, and the areas shaded with light blue and orange are the respective posterior error intervals. The true values of the corresponding quantities are drawn with green line.





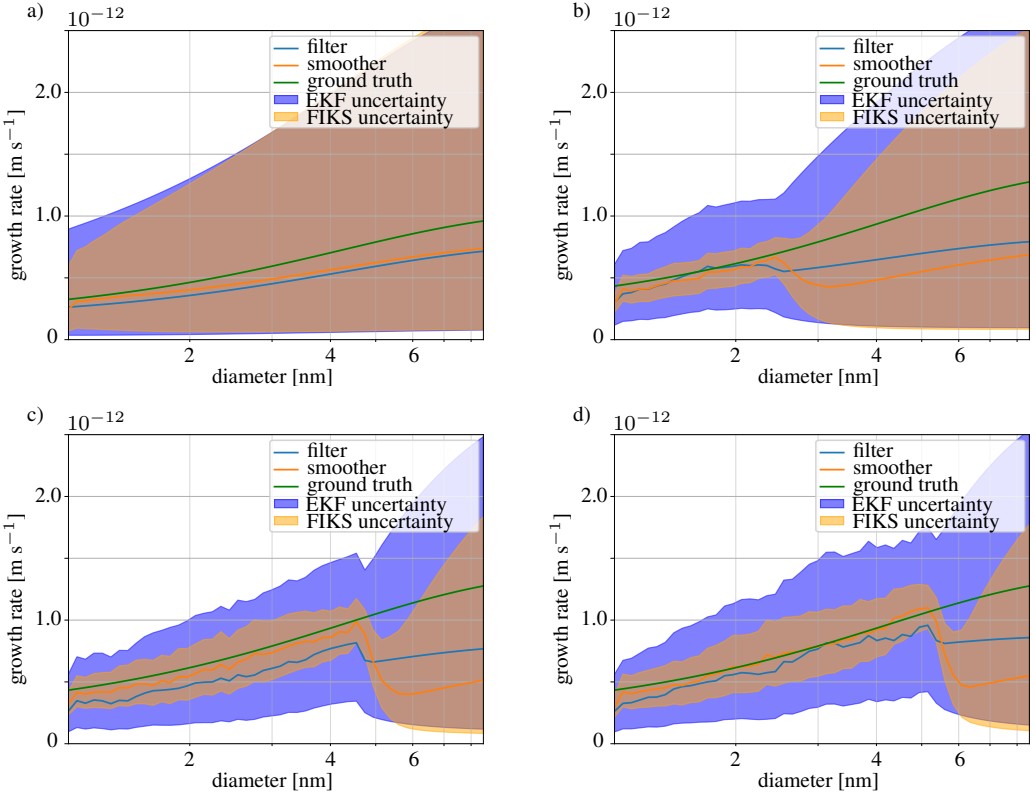

**Figure 7.** Case 3: SS, high SNR. Condensational growth rate estimates corresponding to four instants of time. The blue and orange lines represent the approximate posterior expectations for EKF and FIKS, respectively, and the areas shaded with light blue and orange are the respective posterior error intervals. The true growth rates are marked with green lines.

## 4  Conclusions

The radiative forcing caused by aerosols – and the underlying processes of new particle formation and growth – are currently
understood to be the most uncertain factors in the prediction of climate change. While a significant effort has been directed into experimental campaigns for collecting data that carries indirect information on these processes and their effects, computational methods for analyzing the data are still insufficient for reliable estimation of the process rates.

In this paper, the problem of estimating the nucleation, growth and deposition rates of aerosols was cast in the framework of Bayesian state estimation. We modeled the dynamics of aerosol size distributions with the general dynamics equation, and
considered the (process rate) parameters in the model as unknown state variables. These size- and/or time-dependent variables were estimated together with the particle number density based on sequential particle counter measurements using Extended Kalman filter (EKF) and Fixed interval Kalman smoother (FIKS). Furthermore, to quantify the uncertainties of the estimated variables, we also computed posterior error intervals for the process rate parameters.



**Figure 8.** Case 4: SS, low SNR. State estimates for the particle size density (subfigures a, e and f), nucleation rate (b), growth rate (c), and wall loss rate (d). The images in plots a) and c) depict the approximate, FIKS-based posterior expectations for the entire time-evolutions of the corresponding quantities. Plots e) and f) illustrate the EKF and FIKS estimates for the particle size density corresponding to times 2 h and 10 h, respectively. In plots b) and d)–f), the blue and orange lines represent the approximate posterior expectations for EKF and FIKS, respectively, and the areas shaded with light blue and orange are the respective posterior error intervals. The true values of the corresponding quantities are drawn with green line.



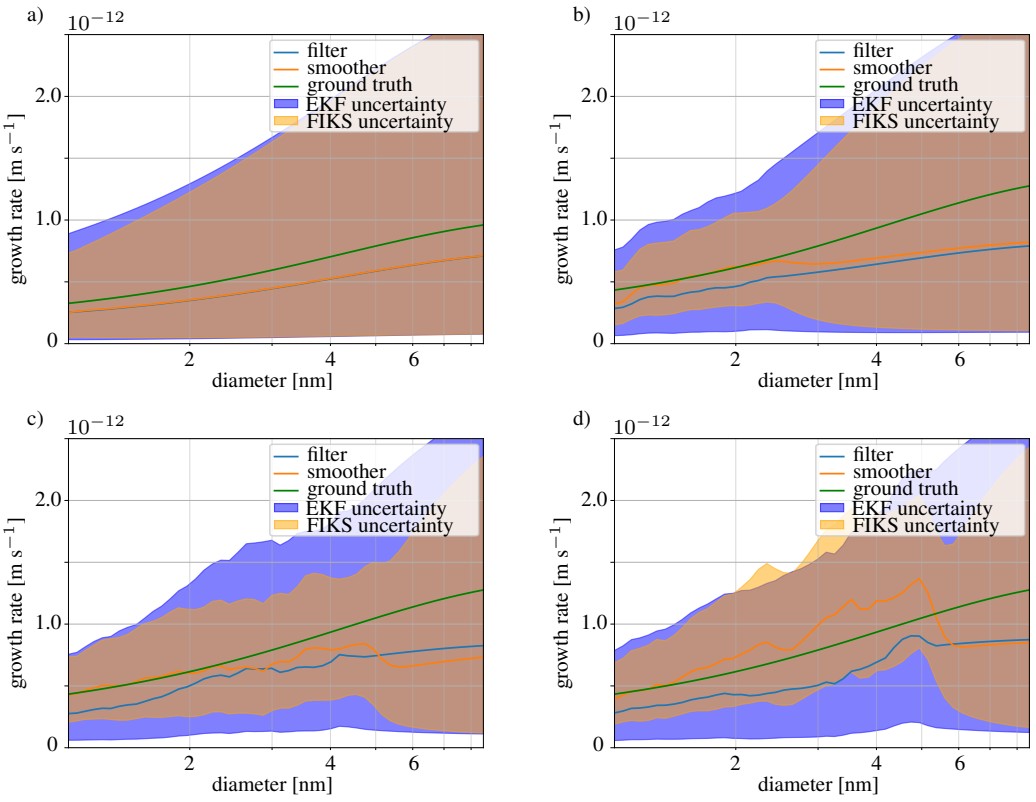

**Figure 9.** Case 4: SS, low SNR. Condensational growth rate estimates corresponding to four instants of time. The blue and orange lines represent the approximate posterior expectations for EKF and FIKS, respectively, and the areas shaded with light blue and orange are the respective posterior error intervals. The true growth rates are marked with green lines.

The approach was tested with a set of numerical simulation studies, where two processes of different types were considered:
1) a nucleation event, which represents qualitatively a typical particle formation event in the atmosphere, and 2) a process approaching steady state, a case, which represents particle formation and growth in a chamber experiment.

The EKF- and especially FIKS-based estimates for the process rates were overall reliable, even in cases of low SNR data. Also the posterior error intervals were feasible, that is, the true process rates lay mostly within the approximate error intervals. The reason why FIKS-estimates are overall superior to EKF-estimates is that Bayesian smoothers use also future data when
estimating a quantity at given time $t$, while filter estimates are based on only the data up to the point $t$. In on-line and control applications, of course, the future data is not available, but in the applications targeted in this paper, the data analysis is done after the experiment, that is, the entire data set is available for Bayesian smoothning. The results of the numerical studies support the feasibility of the proposed approach to estimating the aerosol formation, growth and deposition rates, and quantifying their uncertainties. Based on these findings, we conclude that Bayesian state estimation combined with aerosol particle dynamics
modeling offer a reliable tool for analyzing sequentially measured particle counter data. The estimated aerosol process rates

and their uncertainties can improve the analysis of the experimental data, and provide better insight on the particle formation and growth in the atmosphere. In the future, such analysis can potentially offer improved assessment of the radiative forcing by aerosols and its uncertainty. Eventually, this can lead to improved predictions and uncertainty quantification of the climate change via combination of the enhanced process rate estimates and their uncertainties with global aerosol-climate models.

*Code and data availability.* The current version of the code used to generate the data as well as the implementation of the estimation method will be made available upon publication under the MIT Expat License. The exact version of the code used to produce the results in this paper will be the initial version of the code.

*Author contributions.* The state-space formulation of the aerosol process rate estimation problem was contributed by all authors. MO implemented all software used in the computations. The numerical experiments were planned and analysed by all authors. All authors took active

part in writing the paper.

*Competing interests.* The contact author has declared that neither they nor their co-authors have any competing interests.

## Appendix A:  Parameters in data simulation

Table A1 lists the models and parameters used for simulating the nucleation, condensation and deposition processes, and the particle size density, as well as the modeling the particle counter data.

## Appendix B:  Model details and parameters in state estimation

In this appendix, we describe the choices of models and parameters made for state estimation in the numerical examples (Section 3). We start by summarizing qualitatively the choices of models for the process rates in the two example cases, nucleation event (NE) and steady state (SS), see Table B1.

In the following, we first describe how the smoothness of the size-dependent parameters is formulated. Next, we discuss

modeling the time-dependence of by $1^{st}$ and $2^{nd}$ order Markov models. Finally, we list all parameter values chosen for each test case.

### B0.1  Size-dependent processes, smoothness

The size-dependent process rate variables – deposition rate $\lambda$ in all test cases and the growth rate $g$ in the SS cases – are assumed to be smooth functions of size. Since these variables also depend on time, we model them as multivariate stochastic

processes, particularly, $1^{st}$ order Markov processes, as described below. The smoothness in size is accounted for by modeling





**Table A1.** Summary of the models and parameters used for simulating data in cases of Nucleation Event (NE) and Steady State (SS). Here, the growth rate $g$ is assumed to be a product of a size- and time-dependent parts, that is, $g = g_d(d_{\mathrm{p}})g_t(d_t)$.

| Model/parameter | | Nucleation Event (NE) | Steady State (SS) |
|---|---|---|---|
| $g\,[\mathrm{nm\,h^{-1}}]$ | $g_d(d_{\mathrm{p}})$ | 9 | $5 \cdot \tanh(0.17 \cdot (10^9 \cdot d_{\mathrm{p}} + 0.8))$ |
| | $g_t(d_t)$ | $\begin{cases} \frac{1}{2} \cdot (1 - \cos(2\pi \frac{t-t_0}{t_1-t_0})) & , \ t \in [t_0, t_1] \\ 0 & , \ \text{otherwise} \end{cases}$ | $\begin{cases} \frac{1}{2} \cdot (1 - \cos(\frac{2\pi t}{t_1})) & , \ t \in [0, t_1] \\ 1 & , \ \text{otherwise} \end{cases}$ |
| $\lambda\,[\mathrm{s^{-1}}]$ | | $2.5 \cdot 10^{-4} \left(\frac{r d_{\mathrm{p}}}{d_{\mathrm{p}}^{\min}}\right)^{-\frac{3}{2}} + \frac{5 \cdot 10^{-5}}{1 + \exp\left(-\frac{d_{\mathrm{p}} - d_{\mathrm{p}}^{\min}}{\sigma_d}\right)}$ | $\frac{1.31 \cdot 10^{-12}}{d_{\mathrm{p}}}$ |
| $J\,[\#\mathrm{cm^{-3}s^{-1}}]$ | | $\begin{cases} 20 \cdot (1 - \cos(2\pi \frac{t-t_0}{t_1-t_0})) & , \ t \in [t_0, t_1] \\ 0 & , \ \text{otherwise} \end{cases}$ | $\begin{cases} 50 \cdot (1 - \cos(\frac{2\pi t}{t_1})) & , \ t \in [0, t_1] \\ 1 & , \ \text{otherwise} \end{cases}$ |
| $\beta\,[\#^{-1}\mathrm{cm^3 s^{-1}}]$ | | Seinfeld and Pandis (2016) | Seinfeld and Pandis (2016) |
| GDE (evol. model) | $\Delta_t$ | 3s | 3s |
| | $d_0$ | 13.85nm | 0.87nm |
| | $\frac{d_{i+1}}{d_i}$ | 1.0017 | 1.0014 |
| | $Q$ | 2500 | 1731 |
| Particle counter (obs. model) | $\Delta_t$ | 120s | 120s |
| | $d_0$ | 14.1nm | 1.1nm |
| | $\frac{d_{i+1}}{d_i}$ | 1.0366 | 1.0469 |
| | $M$ | 111 | 50 |
| | $V$ | $0.9, 90\mathrm{cm^3}$ | $2, 200\mathrm{cm^3}$ |

elements of each of the associated random variables (size-discretized process rate variables $\xi_{\varphi}^k = [\xi_{\varphi,1}^k, \ldots, \xi_{\varphi,Q}^k]^{\mathrm{T}}, \ \varphi = \lambda, g$) at a given time $k$ as mutually correlated.

$$\Gamma_{\xi_{\varphi}^k}(i,j) = \begin{cases} \sqrt{\sigma_{\varphi,i}^2}\sqrt{\sigma_{\varphi,j}^2} \exp\left(-\frac{|i-j|}{\delta_\varphi}\right), & \text{for } |i-j| < (Q/2) \\ 0, & \text{for } |i-j| \geq (Q/2) \end{cases} \tag{B1}$$

where $\sigma_{\varphi,i}^2$ is the variance of $\xi_{\varphi,i}^k$ and $\delta_\varphi$ is a parameter defining how steeply the cross-covariance between elements $\xi_{\varphi,i}^k$ and $\xi_{\varphi,j}^k$ decreases as function of the difference between indices $i$ and $j$.


Figure B1 shows the covariance matrices of the two size-dependence variables: deposition rate $\lambda$ in Cases 1 & 2 and growth rate $g$ in Cases 3 & 4. We note here that in Cases 1 & 2, $\sigma_{\lambda,i}^2$ is constant while in Cases 3 & 4, $\sigma_{g,i}^2$ increases with particle size. For details of parameter values, see below.





**Table B1.** Qualitative description of the models used in state estimation for Nucleation Event (NE) and Steady State (SS) cases.

| Process | | NE (Cases 1 & 2) | SS (Cases 3 & 4) |
|---|---|---|---|
| Growth rate $g$ | Size-dependence | - | Smooth |
| | Time-dependence | $2^{nd}$ order Markov process | $1^{st}$ order Markov process |
| Deposition rate $\lambda$ | Size-dependence | Smooth | Smooth |
| | Time-dependence | $1^{st}$ order Markov process | $1^{st}$ order Markov process |
| Nucleation rate $J$ | Size-dependence | - | - |
| | Time-dependence | $2^{nd}$ order Markov process | $1^{st}$ order Markov process |
| Coagulation frequency $\beta$ | Size-dependence | Known | Known |
| | Time-dependence | - | - |

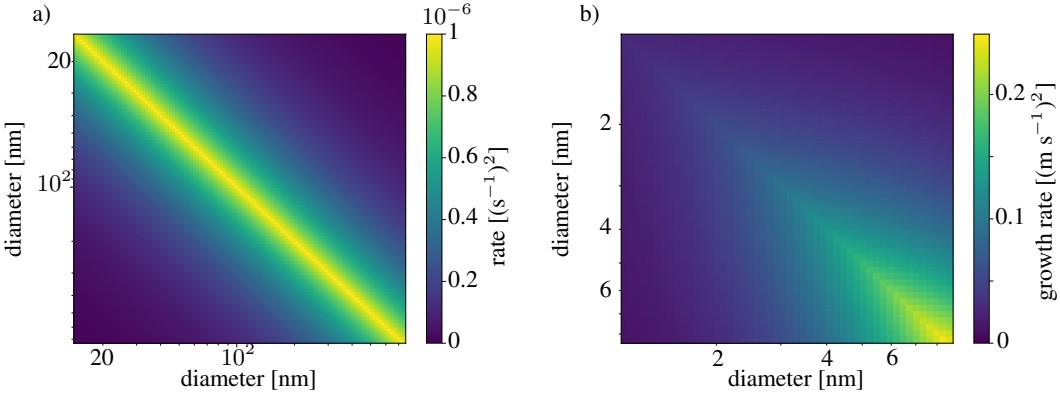

**Figure B1.** Covariance matrices of, a) the deposition rate $\lambda$ in Cases 1 & 2 and b) growth rate $g$ in Cases 3 & 4.

### B0.2   Time-dependence, $1^{st}$ order Markov processes

Assume next that the process rate variable $\xi_\varphi^k$ is modeled as a $1^{st}$ order Markov process. Here, we fix the covariance of $\xi_\varphi^k$ first, as well as the temporal smoothness ($1^{st}$ or $2^{nd}$ order Markov process). Then, we determine the covariance of the driving noise process.

When the covariance matrix $\Gamma_{\xi_\varphi^k}$ of $\xi_\varphi^k$ is time-invariant, we can calculate the covariance matrix of the state noise $\eta_\varphi^k$ using Equation (11) as:

$$\Gamma_{\eta_\varphi^k} = \Gamma_{\xi_\varphi^k} - \Psi_\varphi \Gamma_{\xi_\varphi^k} \Psi_\varphi^{\mathrm{T}}. \tag{B2}$$

In cases of size-dependent variables, we construct the covariance matrix $\Gamma_{\xi_\varphi^k}$ as described above, fix the state transition matrix $\Psi_\varphi = r_\varphi I$, by choosing the parameter $r_\varphi \in ]0,1[$, and finally, compute the state noise covariance matrix using Equation (B2).





In Cases 3 & 4, where also the nucleation rate is modeled as a $1^{\text{st}}$ order Markov process, the procedure is exactly the same. Note, however, that in the case of variable $J$, which does not depend on size, the state-transition matrix and the covariance are simply $\Psi_J = r_J$ and $\Gamma_{\xi_J^k} = \sigma_{J,i}^2$, respectively.

### B0.3 Time-dependence, $2^{\text{nd}}$ order Markov processes

As shown in Table B1, the size-independent variables in Cases 1 & 2 ($g$ and $J$) are modeled as $2^{\text{nd}}$ order Markov processes. For these variables, we choose the root parameters $r_\varphi^1, r_\varphi^2$ in the evolution models by an approach adopted from the analysis of second order systems, such as damped oscillators: We first define a characteristic time is $T_\varphi$ and damping ratio $\zeta_\varphi$, and then calculate $r_\varphi^1$ and $r_\varphi^2$ by solving the set of equations

$$
r_\varphi^1 + r_\varphi^2 = 2\left(1 - \zeta_\varphi \frac{2\pi \Delta t^k}{T_\varphi}\right) \tag{B3}
$$

$$
r_\varphi^1 r_\varphi^2 = 1 - 4\pi \zeta_\varphi \frac{\Delta t^k}{T_\varphi} + 4\pi^2 \left(\frac{\Delta t^k}{T_\varphi}\right)^2. \tag{B4}
$$

These parameters, together with separately chosen variances of the state noises $\eta_g^k$ and $\eta_J^k$ as well as the expectations and variances of the initial states $\xi_g^0$ and $\xi_J^0$ define the properties of the $2^{\text{nd}}$ order Markov models.

### B0.4 Parameter choices

All parameter values chosen for state estimation in each test case are listed in Table B2





**Table B2.** Parameters used in state estimation for Nucleation Event (NE) and Steady State (SS) cases.

| Process | | NE (Cases 1 & 2) | SS (Cases 3 & 4) |
|---|---|---|---|
| Growth rate $g$ | $\xi_g^{0\|0}$ | $0\,\mathrm{m\,s^{-1}}$ | $-2\,2.8\,10^{-13}\,\mathrm{m\,s^{-1}}$ |
| | $\Gamma_{\xi_g}^{0\|0}$ | $(5\,2.8\,10^{-13})^2\,(\mathrm{m\,s^{-1}})^2$ | $(16\,2.8\,10^{-13}\tanh(0.17\cdot(10^9\cdot d_\mathrm{p}+0.8)))^2\,(\mathrm{m\,s^{-1}})^2$ |
| | $\sigma_{\eta g}^2$ | $(5\,2.8\,10^{-13})^2\,(\mathrm{m\,s^{-1}})^2$ | $(8\,2.8\,10^{-13}\tanh(0.17\cdot(10^9\cdot d_\mathrm{p}+0.8)))^2\,(\mathrm{m\,s^{-1}})^2$ |
| | $T_g$ | $1800\,\mathrm{s}$ | $300\,\mathrm{s}$ |
| | $\zeta_g$ | $0.95$ | $0.95$ |
| | $\delta_g$ | not applicable | $50$ |
| Deposition rate $\lambda$ | $\lambda^{0\|0}$ | $6\,10^{-5}\,\mathrm{s}$ | $\frac{1.31\cdot10^{-12}}{d_\mathrm{p}}\,\mathrm{s}$ |
| | $\Gamma_{\xi_\lambda}^{0\|0}$ | $(10^{-3})^2\,\mathrm{s^{-2}}$ | $(0.1\lambda^{0\|0})^2\,\mathrm{s^{-2}}$ |
| | $r_\lambda$ | $1$ | $1$ |
| | $\sigma_{\lambda,i}^2$ | $(10^{-3})^2\,\mathrm{s^{-2}}$ | $(10^{-3})^2\,\mathrm{s^{-2}}$ |
| | $\delta_\lambda$ | $10$ | $10$ |
| Nucleation rate $J$ | $\xi_J^{0\|0}$ | $0\,\mathrm{cm^{-3}s^{-1}}$ | $0\,\mathrm{cm^{-3}s^{-1}}$ |
| | $\Gamma_{\xi_J}^{0\|0}$ | $0.2^2\,(\mathrm{cm^{-3}s^{-1}})^2$ | $10^2\,(\mathrm{cm^{-3}s^{-1}})^2$ |
| | $\sigma_{\eta J}^2$ | $0.2^2\,(\mathrm{cm^{-3}s^{-1}})^2$ | $5^2\,(\mathrm{cm^{-3}s^{-1}})^2$ |
| | $T_J$ | $1800\,\mathrm{s}$ | $300\,\mathrm{s}$ |
| | $\zeta_J$ | $0.95$ | $0.95$ |
| Number density $N$ | $N^{0\|0}$ | $\bar{H}^\mathrm{T}y^1\,\mathrm{cm^{-3}}$ | $0\,\mathrm{cm^{-3}}$ |
| | $\Gamma_N^{0\|0}$ | $4\frac{y^1+100}{V}\,(\mathrm{cm^{-3}})^2$ | $4\frac{y^1+100}{V}\,(\mathrm{cm^{-3}})^2$ |
| | $\Gamma_\epsilon$ | $1\,(\mathrm{cm^{-3}})^2$ | $4\,(\mathrm{cm^{-3}})^2$ |
| Modeling error $\iota^k$ | $\Gamma_\iota$ | $\frac{100}{V}\,(\mathrm{cm^{-3}})^2$ | $\frac{100}{V}\,(\mathrm{cm^{-3}})^2$ |

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
