# Peer review of "Retrieval of process rate parameters in the general dynamic equation for aerosols using Bayesian state estimation: BAYROSOL1.0"

_Geoscientific Model Development, 2020_

## Short Comment (SC1) · 14 Nov 2020

Dear authors,

in my role as Executive editor of GMD, I would like to bring to your attention our Editorial version 1.2:

https://www.geosci-model-dev.net/12/2215/2019/

This highlights some requirements of papers published in GMD, which is also available on the GMD website in the 'Manuscript Types' section:

http://www.geoscientific-model-development.net/submission/manuscript_types.html  >

[Figure]

In particular, please note that for your paper, the following requirements have not been met in the Discussions paper:

- The main paper must give the model name and version number (or other unique identifier) in the title.

- "Code must be published on a persistent public archive with a unique identifier for the exact model version described in the paper or uploaded to the supplement, unless this is impossible for reasons beyond the control of authors. All papers must include a section, at the end of the paper, entitled "Code availability". Here, either instructions for obtaining the code, or the reasons why the code is not available should be clearly stated. It is preferred for the code to be uploaded as a supplement or to be made available at a data repository with an associated DOI (digital object identifier) for the exact model version described in the paper. Alternatively, for established models, there may be an existing means of accessing the code through a particular system. In this case, there must exist a means of permanently accessing the precise model version described in the paper. In some cases, authors may prefer to put models on their own website, or to act as a point of contact for obtaining the code. Given the impermanence of websites and email addresses, this is not encouraged, and authors should consider improving the availability with a more permanent arrangement. Making code available through personal websites or via email contact to the authors is not sufficient. After the paper is accepted the model archive should be updated to include a link to the GMD paper."

Therefore, create a reference to your algorithm and its version in the title of your paper upon revision of the manuscript.

Regarding the code availability section, I want to emphasize, that it is not sufficient to make a proposition to make the code publicly available upon final publication. The code

is part of the review process and thus should be already available for the discussion phase except license issue prevent this. But if you can make it available later on, this seems not to be the case here.

Yours,

Astrid Kerkweg

---

## Referee Comment (RC1) · Anonymous Referee #1 · 21 Nov 2020

This work uses bayesian estimation techniques, generally applied to on-line control problems, to estimate aerosol microphysics from data. These aerosol microphysical processes are important to constrain since they lead to uncertainty in climate change. The estimation technique applied in this work is novel to the field of aerosol microphysics. A filter and smoother is investigated to estimate aerosol nucleation, growth, and loss rates. These methods could be very useful to several experiments and datasets since they not only estimate the microphysical rates but also their uncertainty. I recommend this paper is published given that the following comments are addressed.

General comments:

It seems odd that there is no source term for nucleation in equation (1). I see that it is included as a boundary condition on particle flux in, but even with that it seems like these equations do not correctly represent particle nucleation and growth. Since if the net growth by condensation of $d_p^{min}$ is set to $J$, then either there is no growth of $d_p^{min}$ to larger sizes or the loss of $d_p^{min}$ is included in $J$. Additionally, $J$ is defined as flux of particles ( number of particles per area per time), but then is referenced as particle concentration rate (# $cm^{-3}s^{-1}$) later in the discretized model and numerical simulation. In the discretized model there is in fact a nucleation and growth term for the first bin, so it seems like the error is in the representation of the continuous GDE in equations (1) and (2). This needs to be corrected or clarified. Also, in equation (1) $d_0$ is used as the lower limit integrated over for coagulation sink but $d_0$ is not defined.

It is noted that Case 1 & 2 are set-up to study estimation stability to see if the method can estimate time-invariant wall loss even though the loss rate follows a 1st order Markov model. However, the estimated loss rate is only shown at one time and it is not discussed further. Did the estimated loss rate vary over time, and by how much?

What is the range in SNR between cases 1 and 2 as well as between cases 3 and 4? How is SNR adjused?

The observed difference in estimated and true nucleation rates in cases 3 & 4 is quite interesting. It seems like perhaps the nucleated mass rate matches closer than the nucleated number. Is this the case? If so, it would be interesting to note that FIKS can recover the nucleated mass rate when there are uncertainties in the nucleated particle size.

Minor corrections / suggestions:

1. Line 31-32 "...paying also attention to the uncertainties" is confusing wording. Maybe change to "analyze the data with care and pay attention ..."

2. In general, the citations should have the format (Author1, year1; Author2, year2; ...) unless the citation is a subject in your sentence in which case the format should be just the year in parenthesis, i.e., "this thing was described by Author1 (year1) and Author2 (year2)"

3. GR needs to be defined as growth rate when it is introduced in line 38.

4. Citations are repeated in the paragraph starting at line 60. This needs to be fixed.

5. Line 70 change to "the Bayesian approach was adopted to estimate aerosol size distributions"

6. Line 182 - describe what the notation $]0, 1[$ means. I am used to seeing $x \in (0, 1)$ for $0 < x < 1$ and $x \in [0, 1]$ for $0 \leq x \leq 1$

7. It would be helpful to explicitly describe what $\bar{y}^k$ and $z^k$ represent (number of particles counted?).

8. Figures 7a - 7d need timestamps.

9. Reference Appendix B in the text (near lines 180-190) to describe how $r_\varphi$ is chosen.

10. In algorithm 2, it seems like it should be a loop over $k = K - 1, ..., 1$ or there should be a separate case for if $k = K$ since it is not clear that $\Gamma^{K+1|K}$ or $X^{K+1|K}$ exist.

11. Figures 2a and 2b look very similar to my eyes. It would be nice to show the surface plot of their difference, potentially instead of the current figure 2b.

---

## Author Comment (AC1) · 8 Feb 2021

**Retrieval of process rate parameters in the general dynamic equation for aerosols using Bayesian state estimation of aerosol dynamics: BAYROSOL1.0**

Matthew Ozon, Aku Seppänen, Jari P. Kaipio, Kari E.J. Lehtinen

**1 Thanks**

We thank the reviewer for his/her thoughtful comments and will revise our manuscript accordingly. In the following are our point-by-point responses to the reviewer's remarks in such a way that we have listed the reviewer's remarks in blue and our reply is in black font.

**2 General comments**

**2.1 Paragraph 1**

It seems odd that there is no source term for nucleation in equation (1). I see that it is included as a boundary condition on particle flux in, but even with that it seems like these equations do not correctly represent particle nucleation and growth. Since if the net growth by condensation of $d_p^{min}$ is set to $J$, then either there is no growth of $d_p^{min}$ to larger sizes or the loss of $d_p^{min}$ is included in $J$.

The reviewer is correct in noting that in equation 1 there is no term for nucleation and we have explained our approach insufficiently. Our approach here considers particle dynamics above the size range in which the actual nucleation occurs, i.e. a typical DMPS or SMPS measurement range. Then, the appearance of new particles to the lowest end of the measurement range occurs through condensation from even smaller particles that have nucleated slightly earlier. This process has sometimes been called "apparent particle formation", and we adopt this terminology here to avoid further confusion. Such a growth of particles across the lower limit of our particle size range is most conveniently treated mathematically as a boundary condition, resulting in the discretized model (equation 5) as a source term for the lowest size bin. Thus, in our revision we will replace "nucleation" with "apparent particle formation" everywhere and add on page 4 (before equation 2): "*We do not include an explicit nucleation term in equation 1 as we are considering a size range typical for particle mobility (DMPS*

*or SMPS) measurements which is above nucleation size. Then, appearance of new particles to the measurement range occurs by condensational growth of freshly nucleated particles from below measurement range. This process is sometimes called apparent particle formation (e.g. Lehtinen et al., 2007) and mathematically it is conveniently treated as a particle concentration flux in size space ($cm^{-3}s^{-1}$) boundary condition for the GDE."* This flux term can be written as a product $g \times n$ (equation 2), signifying that it is affected only by the condensation. This, however, does not mean that the deposition is neglected in the smallest size class.

Additionally, $J$ is defined as flux of particles (number of particles per area per time), but then is referenced as particle concentration rate (# $cm^{-3}s^{-1}$) later in the discretized model and numerical simulation. In the discretized model there is in fact a nucleation and growth term for the first bin, so it seems like the error is in the representation of the continuous GDE in equations (1) and (2). This needs to be corrected or clarified. Also, in equation (1) $d_0$ is used as the lower limit integrated over for coagulation sink but $d_0$ is not defined.

This comment by the reviewer likely stems from our poor explanation of the treatment of nucleation (or apparent particle formation, see our reply to previous comment). The discretization of the GDE with apparent particle formation rate as a boundary condition results directly in the discretized equations 5 and 6, so no consistency problem there. We agree with the reviewer that we use the term flux non-traditionally as the propagation of particle concentration in our case occurs in particle diameter space (and not "normal" space). This is clarified by the addition mentioned in our response to the previous comment: *"We do not include an explicit nucleation term in equation 1 as we are considering a size range typical for particle mobility (DMPS or SMPS) measurements which is above nucleation size. Then, appearance of new particles to the measurement range occurs by condensational growth of freshly nucleated particles from below measurement range. This process is sometimes called apparent particle formation (e.g. Lehtinen et al., 2007) and mathematically it is conveniently treated as a particle concentration flux in size space ($cm^{-3}s^{-1}$) boundary condition for the GDE"* In the revised version, we will also define $d_0$ clearly. It is the actual (physical) diameter at which the nucleation occurs. Moreover, we have highlighted the difference between $d_0$ and $d_p^{\min}$ – the latter of which is the "apparent nucleation size", which represents the smallest diameter in the model. Also, we will change the coagulation source in equation (1) since the integral range should start from the same size as the coagulation loss, $d_0$ instead of 0.

**2.2 Paragraph 2**

It is noted that Case 1 & 2 are set-up to study estimation stability to see if the method can estimate time-invariant wall loss even though the loss rate follows a 1st order Markov model. However, the estimated loss rate is only shown at one time and it is not discussed further. Did the estimated loss rate vary over time, and by how much?

Yes, the wall loss was treated as time varying, even if it is expected to be time invariant (but size dependent). The Extended Kalman filter (EKF) results for the wall loss rates showed time dependency while the Kalman smoother (FIKS) produced very weak time dependence. See some results at different times in fig. 1. We choose not to show these figures in the

[Figure]

Figure 1: Wall loss estimation for four different instants in time.

manuscript but will add a sentence on this in the revised discussion.

**2.3 Paragraph 3**

What is the range in SNR between cases 1 and 2 as well as between cases 3 and 4? How is SNR adjused?

The SNR was adjusted by choosing the sample volume ($V$, via the detector-sample-flow rate $\phi_a$) of the CPC. As noted in Section 3.1.1, the SNR of CPC data increases with $V$. The associated Poisson distributed noise is the main source of noise also in the real experiment, and choosing

$V$ is a trade-off between SNR and the duration of the measurement. In the studied cases, the SNR is controlled by $\phi_a$ so that the time base of the measurements is the same between different SNRs. The actual ranges for the SNRs are case 1 $[0, 6426]$, case 2 $[0, 64.26]$, case 3 $[0, 4440]$ and case 4 $[0, 44.4]$, and we will mention them in the respective sections.

**2.4 Paragraph 4**

The observed difference in estimated and true nucleation rates in cases 3 & 4 is quite interesting. It seems like perhaps the nucleated mass rate matches closer than the nucleated number. Is this the case? If so, it would be interesting to note that FIKS can recover the nucleated mass rate when there are uncertainties in the nucleated particle size.

The underestimation of nucleation rates in cases 3 and 4 resulted from the fact that we were comparing particle formation rates at two different sizes, 0.87 nm and 1.1 nm. The rates at 1.1 nm are lower because as the particles grow from 0.87 nm to 1.1 nm their concentration is decreased by (mainly) deposition onto the walls. We have now corrected our analysis and in figure 2 one finds the new versions of figures 6b and 8b, showing an excellent match for the predicted particle formation rates. We also change the text (on page 19) accordingly.

[Figure]

[Figure]

Figure 2: Adding the true (simulated) particle flux at 1.1 nm (size at which the nucleation is estimated by the method), a.k.a. apparent nucleation rate.

**3 Minor corrections and suggestions**

1 Line 31-32 "...paying also attention to the uncertainties" is confusing wording. Maybe change to "analyze the data with care and pay attention ..." We will revise this as suggested.

2 In general, the citations should have the format (Author1, year1; Author2, year2; ...) unless the citation is a subject in your sentence in

which case the format should be just the year in parenthesis, i.e., "this thing was described by Author1 (year1) and Author2 (year2)" We will revise the citations according to the guidelines of the journal.

3 GR needs to be defined as growth rate when it is introduced in line 38. This will be corrected in the revised version of the manuscript.

4 Citations are repeated in the paragraph starting at line 60. This needs to be fixed. This will be corrected in the revised version of the manuscript.

5 Line 70 change to "the Bayesian approach was adopted to estimate aerosol size distributions" This will be corrected in the revised version of the manuscript.

6 Line 182 - describe what the notation $]0, 1[$ means. I am used to seeing $x \in (0, 1)$ for $0 < x < 1$ and $x \in [0, 1]$ for $0 \leqslant x \leqslant 1$

7 It would be helpful to explicitly describe what $\tilde{y}_i^k$ and $z_i^k$ represent (number of particles counted?). $z_i^k$ is defined in equation (16) and the line above it and $\tilde{y}^k$ is the number of particles counted by the CPC, which we add after equation (17)

8 Figures 7a - 7d need timestamps. Will be done. And the same will be applied to figures 9a – 9d.

9 Reference Appendix B in the text (near lines 180-190) to describe how $r_\phi$ is chosen. This will be corrected in the revised version of the manuscript.

10 In algorithm 2, it seems like it should be a loop over $k = K - 1, \ldots, 1$ or there should be a separate case for if $k = K$ since it is not clear that $\Gamma^{K+1|K}$ or $X^{K+1|K}$ exist. Thank you for noticing this mistake. Will be corrected.

11 Figures 2a and 2b look very similar to my eyes. It would be nice to show the surface plot of their difference, potentially instead of the current figure 2b. The difference in the transfer functions is resolution, which is illustrated in the panel c) of figure 3 for one channel. The transfer function used in the method corresponds to the average of the "true" transfer function over each discretization bins, hence the difference in amplitude between the fine and coarse models. It is clearly illustrated in the panel c) of figure 3 where the maximum value of the averaged model, in orange, is smaller than that of the fine model, in green — which is merely the evaluation of transfer function, and not the average. The figure 3 will replace figure 2 in the manuscript.

[Figure]

Figure 3: New transfer/kernel function plot.

**Additional reference**

Lehtinen, K. E. J., Dal Maso, M., Kulmala, M. and Kerminen, V.-M. (2007) Estimating nucleation rates from apparent particle formation rates and vice versa: Revised formulation of the Kerminen-Kulmala equation. Journal of Aerosol Science, Vol. 38, No. 9, 2007, p. 988-994.

---

## Referee Comment (RC2) · Anonymous Referee #2 · 16 Feb 2021

The study proposes an inverse modeling approach, based on Bayesian state estimation, to assess the parameters controlling the evolution of an atmospheric aerosol particle size distribution during particle formation and growth, as well as the parameter uncertainties. The model is applicable to measured particle number concentration data, plenty of which is available from field and laboratory measurements of particle formation events.

The application of different state estimation techniques, namely the Extended Kalman filter (EKF) and the Fixed Interval Kalman Smoother (FIKS) is demonstrated using model-generated synthetic particle distribution data. The results seem very promising:

[Figure]

the "true" parameters used for the data generation are generally well recovered, and especially the FIKS smoother reduces the uncertainty ranges of the key parameters. This is an important step forward in aerosol data analysis, since traditionally the parameters are assessed applying simple assumptions with no sophisticated uncertainty estimates.

In addition to contributing to improving the understanding of aerosol microphysics and processes controlling atmospheric aerosol distributions, the information that the proposed methods give can also improve the description of aerosols in larger-scale transport and climate models. The work is thus in the scope of Geoscientific Model Development. I have a few comments that I would ask the authors to address before recommending the paper for publication:

1. If I understand correctly, no constraints for the fitted parameters are used in the preset work. Could a measurement of the gas-phase condensable vapors provide a useful upper-limit constraint for the condensation rate parameter g (corresponding to irreversible condensation)? Would this improve the results? Similarly, the upper limit of the nucleation rate could possibly be assessed if the identities of the nucleating vapors are reasonably well known (at least in well-controlled laboratory experiments).

2. Figure 1: A minor observation on the NE event: the pre-existing distribution of larger aerosols seems to persist throughout the event, although normally atmospheric observations show that it is diluted due to boundary layer growth around noon (thus also enhancing the particle formation efficiency). Can such a dilution effect be added to the Bayesian model equations?

3. P7, L173: It is reasoned that all the unknown parameters of the GDE are non-negative. However, isn't it possible that particles shrink at low vapor concentrations

(e.g. Salma et al., Atmos. Chem. Phys., 16, 7837-7851, 2016), thus making the condensational growth rate g negative? (Similarly, also the formation rate J may be negative in the case of shrinking particles.)

4. Is it controlled that the applied Euler's method for time integration does not contain (cumulative) errors? While the integration is stabilized by the CFL criterion, does it ensure that a shorter time step will not have a notable effect on the result? Short steps may become relevant at high particle concentrations in polluted atmospheric environments, such as very polluted urban conditions.

5. While previously introduced GDE-based methods to assess particle growth rates or other parameters (i.e. those cited on P3, L60-67, and also the more recent methods proposed by Pichelstorfer et al., Atmos. Chem. Phys., 18, 1307-1323, 2018) do not include parameter uncertainty estimates, can it still be useful to test also these methods against the Bayesian state estimation?

6. It is explained that the evolution of the aerosol size distribution is determined by the nucleation, condensation, coagulation, and sink parameters. I'm wondering if the aerosol particles can exist in different charging states, and if this can have effects on the distribution through e.g. modified growth and coagulation kernels?

7. P4, L93-94: It is stated that the formation of particles occurs typically at 1.5-2 nm. Yet, the nucleation size in the simulations is assumed to be only 0.87 nm, which is barely a molecular size and would generally be expected to correspond to "pre-nucleation" molecular clusters. How was this size chosen? Are the results affected if the size is erroneously assigned?

8. Furthermore, for the steady state (SS) simulation case, the modeled size range is divided to 1731 logarithmically distributed size bins in the range [0.87 nm, 10.00 nm] (P16, L383). For such a high number of bins in a narrow range, the bins are very dense especially at the smallest particle sizes.

Is it ensured that the width is still physically consistent, that is, corresponds to at least one-molecule increment in terms of particle growth? I believe that for large atmospheric molecules, such as oxidized organic species, 10 nm-particles may consist of significantly less than 1731 molecules, in which case the bin widths would be unphysical.

9. P4, L96-97: It is stated that the particle flux to the smallest measurable size class is driven by condensational growth, as defined by Eq. (2). Can coagulation also play a role in polluted environments with high concentrations of nucleated particles? Would this affect the model formulation?

10. P16, L361 onward: Why is a wall loss rate estimate discussed for the "nucleation event" (NE) case? It is explained that the NE case represents a typical particle formation event in the atmosphere. Which type of walls in the ambient atmosphere does the loss rate correspond to, or does it refer to the walls inside a measurement device?

Technical comments:

1. The colored solid lines, corresponding to the "true" and estimated parameters as functions of time or particle size, in most figures are somewhat difficult to see; can they be made e.g. thicker and brighter (and maybe have different line styles)?

[Figure]

Also, the shaded areas corresponding to the uncertainties in the estimates could preferably be lighter and/or more transparent in order to see the lines better. It may also not be obvious by first look that the brown shade is the overlap between the blue and yellow shades; the figure legends list blue and yellow shades, but in most figures the yellow is almost entirely missing and there is only brown / blue.

2. P2, L42: Please remove the comma in "Methods 1 and 2 are applicable to cases, in which..."

---

## Author Comment (AC2) · 24 Feb 2021

**Retrieval of process rate parameters in the general dynamic equation for aerosols using Bayesian state estimation of aerosol dynamics: BAYROSOL1.0**

Matthew Ozon, Aku Seppänen, Jari P. Kaipio, Kari E.J. Lehtinen

**1   Thanks**

We thank the reviewer for his/her thoughtful comments and will revise our manuscript accordingly. In the following are our point-by-point responses to the reviewer's remarks in such a way that we have listed the reviewer's remarks in blue and our reply is in black font.

**2   Comments**

**2.1   Comment 1**

If I understand correctly, no constraints for the fitted parameters are used in the preset work. Could a measurement of the gas-phase condensable vapors provide a useful upper-limit constraint for the condensation rate parameter g (corresponding to irreversible condensation)? Would this improve the results? Similarly, the upper limit of the nucleation rate could possibly be assessed if the identities of the nucleating vapors are reasonably well known (at least in well-controlled laboratory experiments).

The positivity constraint is the only constraint applied. If additional physical information exists, e.g. vapor concentrations as the reviewer suggests, this can be straightforwardly incorporated in the estimations. And, as a general rule, the more information available, the better the estimation.

**2.2   Comment 2**

Figure 1: A minor observation on the NE event: the pre-existing distribution of larger aerosols seems to persist throughout the event, although normally atmospheric observations show that it is diluted due to boundary layer growth around noon (thus also enhancing the particle formation efficiency). Can such a dilution effect be added to the Bayesian model equations?

Yes, dilution can be straightforwardly added to the model, as can be any process that can be modeled with a time evolution equation.

**2.3 Comment 3**

P7, L173: It is reasoned that all the unknown parameters of the GDE are non-negative. However, isn't it possible that particles shrink at low vapor concentrations (e.g. Salma et al., Atmos. Chem. Phys., 16, 7837-7851, 2016), thus making the condensational growth rate g negative? (Similarly, also the formation rate J may be negative in the case of shrinking particles.)

Yes, an aerosol can also evaporate, causing a negative growth rate. And as the reviewer points out, this makes also a negative apparent particle formation rate possible. The choices made in this paper (the parametrization, i.e., the softplus variable transform in Equation (10) and the way we discretize the GDE in Equations (5)-(6)) imply that the growth and formation rates are positive. However, the adaptation to cases where aerosols evaporate is straightforward.

We add the following sentences in the revised manuscript, after equation (6): "The choice for approximating the derivative in the growth term in the discretization of the GDE is made here assuming that the particle growth rate is positive. The modification to cases where the aerosols evaporate, i.e., where growth rate is negative, is straightforward."

**2.4 Comment 4**

Is it controlled that the applied Euler's method for time integration does not contain (cumulative) errors? While the integration is stabilized by the CFL criterion, does it ensure that a shorter time step will not have a notable effect on the result? Short steps may become relevant at high particle concentrations in polluted atmospheric environments, such as very polluted urban conditions.

We acknowledge that the explicit Euler's method used for time integration is not the most accurate one. However, the adoption of any other time integration scheme (such as implicit Euler or Crank-Nicholson) is straightforward, and only affects the evolution model. On the other hand, in cases where the state-space systems are identifiable, the state estimates are generally tolerant to inaccuracies in the evolution model: the sequential measurement data is used for correcting the inaccurate predictions by the evolution model at every time step, and this prevents the cumulation of the time discretization error. We also note that the tolerance with respect to such modeling errors can be further improved by so-called approximation error analysis J. Huttunen and J. Kaipio (1). We will add a sentence on this topic also to the manuscript.

**2.5 Comment 5**

While previously introduced GDE-based methods to assess particle growth rates or other parameters (i.e. those cited on P3, L60-67, and also the more

recent methods proposed by Pichelstorfer et al., Atmos. Chem. Phys., 18, 1307-1323, 2018) do not include parameter uncertainty estimates, can it still be useful to test also these methods against the Bayesian state estimation?

We have chosen here not to compare against other methods as the use of simulated measurement data enables us to compare with the true 'answers'. BAYROSOL is, however, compared against just the methodology mentioned by the reviewer Pichelstorfer et al. (2) (Pichelsdorfer et al., 2018) in a forthcoming paper (submitted to Atmospheric Chemistry and Physics) where the method is applied to experimental aerosol size distribution evolution data obtained at CLOUD/CERN. The methods agree very well, however, as mentioned, BAYROSOL also provides uncertainty estimates for the parameters estimated.

**2.6   Comment 6**

It is explained that the evolution of the aerosol size distribution is determined by the nucleation, condensation, coagulation, and sink parameters. I'm wondering if the aerosol particles can exist in different charging states, and if this can have effects on the distribution through e.g. modified growth and coagulation kernels?

Yes, different charging states are possible, as suggested by the reviewer, and charges will definitely affect the dynamics. Here, in our first tests of BAYROSOL, we have, however, started from the simplest case — neutral particles. It is, however, conceivable that the methodology applies well also to an aerosol population with several charging states if measurement of both size and charge distributions of the charged particles are available. Then, of course, GDEs for the charged particles are needed, as e.g. described in Leppä et al. (3).

**2.7   Comment 7**

P4, L93-94: It is stated that the formation of particles occurs typically at 1.5-2 nm. Yet, the nucleation size in the simulations is assumed to be only 0.87 nm, which is barely a molecular size and would generally be expected to correspond to "pre-nucleation" molecular clusters. How was this size chosen? Are the results affected if the size is erroneously assigned?

The size 0.87 nm was chosen arbitrarily, and as the reviewer remarks, is unphysically small if atmospheric nucleation is concerned. As the purpose of this manuscript, however, concerning nucleation is to test how well the nucleation rate is estimated, the actual choice of nucleation size does not affect any of the results concerning method performance.

**2.8   Comment 8**

Furthermore, for the steady state (SS) simulation case, the modeled size range is divided to 1731 logarithmically distributed size bins in the range [0.87 nm, 10.00 nm] (P16, L383). For such a high number of bins in a narrow range, the bins are very dense especially at the smallest particle sizes.

Is it ensured that the width is still physically consistent, that is, corresponds to at least one-molecule increment in terms of particle growth? I believe that for large atmospheric molecules, such as oxidized organic species, 10 nm-particles may consist of significantly less than 1731 molecules, in which case the bin widths would be unphysical.

The reviewer is correct in stating that the bin width, especially at the lower end of the size spectrum, is unphysically narrow if compared with molecular size. Mathematically, this is, however, no problem as we are numerically solving the continuous form of the GDE.

**2.9   Comment 9**

P4, L96-97: It is stated that the particle flux to the smallest measurable size class is driven by condensational growth, as defined by Eq. (2). Can coagulation also play a role in polluted environments with high concentrations of nucleated particles? Would this affect the model formulation?

Yes, in the most polluted environments the particle number concentrations can be so high that coagulation can play an important role in the particle flux into the measurable size range. In such case, the boundary condition described by Equation (2) may no longer be valid. However, in this case, we can include the flux of particles caused by coagulation into term J, and the form of the discretized GDE will remain unchanged.

**2.10   Comment 10**

P16, L361 onward: Why is a wall loss rate estimate discussed for the "nucleation event" (NE) case? It is explained that the NE case represents a typical particle formation event in the atmosphere. Which type of walls in the ambient atmosphere does the loss rate correspond to, or does it refer to the walls inside a measurement device?

What is meant by wall loss is actually a deposition loss, e.g. sedimentation. We will make use of the term losses instead of wall losses to avoid any further confusion. The measurement model does not account for losses in the tubing of the device, though it can be straightforwardly modified to take it into account if the loss rates are known.

**3 Technical comments**

**3.1 Comment 1**

The colored solid lines, corresponding to the "true" and estimated parameters as functions of time or particle size, in most figures are somewhat difficult to see; can they be made e.g. thicker and brighter (and maybe have different line styles)?

In the revised version of the manuscript, we will increase the linewidth parameter of the plots to thicken the curves.

Also, the shaded areas corresponding to the uncertainties in the estimates could preferably be lighter and/or more transparent in order to see the lines better. It may also not be obvious by first look that the brown shade is the overlap between the blue and yellow shades; the figure legends list blue and yellow shades, but in most figures the yellow is almost entirely missing and there is only brown / blue.

The figure will be modified to make the blue and orange shaded area more transparent.

**3.2 Comment 2**

P2, L42: Please remove the comma in "Methods 1 and 2 are applicable to cases, in which. . ."

We will modify the punctuation to restore the proper grammatical sense of this sentence.

**References**

[1] J. Huttunen and J. Kaipio *Approximation error analysis in nonlinear state estimation with an application to state-space identification*, Inverse Problems, 23, 5, p 2141, 2007.

[2] Pichelstorfer et al. *Resolving nanoparticle growth mechanisms from size- and time-dependent growth rate analysis*, Atmospheric Chemistry and Physics, 18, 2, 1307-1323, 2018.

[3] J. Leppä et al. *Ion-UHMA: a model for simulating the dynamics of neutral and charged aerosol particles*, Boreal Environment Research, 14, 559, 2009